



# Evidence for high-elevation salar recharge and interbasin groundwater flow in the Western Cordillera of the Peruvian Andes

Odiney Alvarez-Campos[1], Elizabeth J. Olson[1], Marty D. Frisbee[1], Sebastián A. Zuñiga Medina[2], José Díaz Rodríguez[2], Wendy R. Roque Quispe[3], Carol I. Salazar Mamani[3], Midhuar R. Arenas Carrión[3], Juan Manuel Jara[3], Alexander Ccanccapa-Cartagena[4,5], Chad T. Jafvert[4,6], and Lisa R. Welp[1]

[1]Earth, Atmospheric, and Planetary Sciences, Purdue University, West Lafayette, Indiana, 47906, USA
[2]Departamento de Geología, Geofísica y Minas, Universidad Nacional de San Agustín de Arequipa, Arequipa, Perú
[3]Departamento de Ingeniería Ambiental, Universidad Nacional de San Agustín de Arequipa, Arequipa, Perú
[4]Lyles School of Civil Engineering, Purdue University, West Lafayette, IN, 47907, USA
[5]Escuela Profesional de Antropología, Universidad Nacional de San Agustín de Arequipa, Av. Venezuela S/N, 04000, Arequipa, Perú
[6]Division of Environmental and Ecological Engineering, Purdue University, West Lafayette, Indiana, 47907, USA

*Correspondence to*: Lisa R. Welp (lwelp@purdue.edu) or Marty D. Frisbee (mdfrisbee@purdue.edu)

**Abstract.** Improving our understanding of hydrogeological processes on the western flank of the central Andes is critical to communities living in this arid region. Groundwater emerging as springs at low elevations provides water for drinking, agriculture, and baseflow. Some springs also have recreation or religious significance. However, the high elevation sources of recharge and specific groundwater flowpaths that support these springs and convey groundwater to lower elevations in southern Peru remain poorly quantified in this geologically complex environment. The objectives of this study were to identify recharge zones and groundwater flowpaths supporting natural springs east of the city of Arequipa in the volcanic mountain terrain, particularly, the potential for recharge within the high-elevation closed-basin Lagunas Salinas salar. We used geochemical and isotopic tracers in springs, surface waters (rivers and lakes), and precipitation (rain and snow) sampled from March 2019 through February 2020. We obtained monthly samples from six springs, bimonthly samples from four rivers, and various samples from high-elevation springs during the dry season. We analyzed stable water isotopes ($\delta^{18}O$ and $\delta^2H$) and general chemistry of springs, rivers, local rainfall, and snow from Pichu Pichu volcano. The monthly isotopic composition of spring water was invariable over time, suggesting that the springs receive a stable source of groundwater recharge and are not supported by relatively short groundwater flowpaths. The chemistry of springs in the low- and mid-elevations (2500 to 2900 masl) point towards a mix of recharge from the salar (4300 masl) and mountain-block recharge (MBR) in or above a queñuales forest ecosystem at ~4000 masl on the adjacent Pichu Pichu volcano. Springs at higher elevation closer to the salar and in a region with a high degree of faulting had higher chloride concentrations indicating higher proportions of interbasin groundwater flow from the salar. We conclude that while the salar is a closed basin, surface water from the salar recharges through the lacustrine sediments, mixes with mountain-block groundwater, and is incorporated into the regional groundwater flow system. Groundwater flow in the mountain block and the subsequent interbasin groundwater flow is accommodated



through extensive faulting and fracturing. Our findings provide valuable information on the flowpaths and zones of recharge that support low-elevation springs in this arid region. In this study, high-elevation forests and a closed-basin salar are important sources of recharge. These features should be carefully managed to prevent impacts to the down-valley springs and streams.

## 1 Introduction

Predicted changes in climate including increases in temperature and evapotranspiration and changes in precipitation patterns in the tropical Andes (Urrutia and Vuille, 2009; Somers et al., 2019) are projected to have a negative impact on the long-term sustainability of critical groundwater resources in southern Peru (Vuille et al., 2018). High-elevation zones of groundwater recharge in the Andes are extremely sensitive to changing climate and anthropogenic land-use change altering vegetative land cover, energy balance, and water balance. This is particularly problematic since groundwater recharge is often enhanced at high elevations due to the delicate interplay between evapotranspiration and precipitation with increasing elevation. Recharge occurring in the mountain block at high elevations is called mountain-block recharge (MBR; see Manning and Solomon, 2003; Wilson and Guan, 2004; Wahi et al., 2008; Ajami et al., 2011; Bresciani et al., 2018). In high-elevation recharge zones, if the amount of precipitation decreases, then the amount of effective precipitation (precipitation minus evapotranspiration) available for MBR likewise decreases. In addition, if the land cover changes in the recharge zone (*e.g.,* changes in vegetation type, density, health and/or movement of the treeline), then the MBR will also change. While the effects of reduction in recharge may not be immediately felt in the region, they are not inconsequential. Groundwater flow within the mountain block, across the mountain front, and to the adjacent valleys will ultimately be affected. This impacts spring flow, headwater streams that originate in the mountains and flow across the mountain front, and groundwater wells often located at lower elevations along the mountain front or in adjacent valleys.

Springs emerging within and down-gradient from the mountain block, for example, are important in the high, arid Andes of Peru for a variety of reasons. They provide a source of potable water and can be used for irrigation and recreation, and some have ecological or religious significance. Communities at lower elevations of the western Andes receive very low annual precipitation and rely on surface runoff and groundwater originating from higher elevations. Consequently, identifying groundwater recharge zones and source areas for springs can help us better understand the potential impact of climate change to water resources and better inform water management plans including the protection of perennial springs and their high-elevation recharge zones. This is complicated by the presence of closed-basin salars located at high elevations in southern Peru. Some of the runoff that occurs during the wet season is captured by these basins and does not contribute (at least directly) to surface runoff across the mountain front. Furthermore, these basins are not usually considered a recharge zone.



Within the high elevations of western South America, the dry climate and unique topography has formed the world's greatest volume of salars. There are more than one hundred salars in topographic depressions in the high mountainous arid region of the Central Andes (Warren, 2016). These salars occur within a tectonically active region with largely interior (endorheic)

drainage from the Andean highland plateaus of Chile, Bolivia, Argentina, and southern Peru. Endorheic basin salars form because the intensity of rainfall is spatially and seasonally variable, the topography of the region favors water accumulation in depressions, and evaporation rates are very high (Warren, 2016). Salars are generally considered closed basins, but not all salars are terminal lakes (Warren, 2016; Risacher et al., 2003). This means that salar brines can infiltrate, be transported elsewhere, and/or mix with meteoric waters (Warren, 2016; Risacher et al., 2003). By using the ratio of the total mass of a

conservative component (*e.g.,* chloride) in the brine lakes to the annual input flux to estimate the component's residence time, Risacher et al. (2003) determined that the residence times of $Cl^-$ in Chilean brine lakes were short (few to hundreds of years) and that such lakes are in flow-through steady-state conditions. This means that eventually most dissolved salts entering salars in this region are lost by leakage through bottom sediments and reenter the hydrologic system (Risacher et al., 2003), which suggests that salars may play an important role as sources of groundwater recharge. In general, the potential for high-elevation

closed-basin salars, which are commonly thought to represent the hydrological dead-end of surface flowpaths, to contribute groundwater recharge to support springs in the central Andes remains uncertain. This begs the question; are all high-elevation salars simply evaporation pans or can leakage beneath the salar contribute to groundwater recharge?

Hydrogeologic research has been conducted in salars and groundwaters of Chile to understand groundwater recharge processes,

sources of spring flow, and the hydrogeologic connectivity of high-elevation mountains and salar basins to the down valley aquifers (Jayne et al., 2016, Scheiling et al., 2017, Herrera et al., 2016; Fritz et al., 1981). Groundwater stable isotopes depleted in deuterium ($^2H$) and oxygen-18 ($^{18}O$) suggest that the main sources of aquifer recharge consist of precipitation at high elevations (~4000 m.a.s.l) of the Altiplano and Andean mountain ridge (Jayne et al. 2016, Scheiling et al., 2017). Early stable isotope research proposed that groundwater recharge to the Pampa del Tamarugal (~1000 m.a.s.l.), which is the most

economically important aquifer in the Atacama Desert, occurs in the nearby Precordillera Altos de Pica (~3500 m m.a.s.l.; Fritz et al., 1981). Subsequent studies pointed to recharge further inland at the Salar de Huasco basin (~3700 m.a.s.l.) through deep fissures that connect the salt-crusted salar to springs in the town of Pica and the Pampa de Tamarugal (Magaritz et al. 1989, 1990). However, recent research rejected the possible influence of the high elevation Salar de Huasco to groundwater recharge of lower elevation springs based on isotopic characterization (Uribe et al., 2015) and geothermometer inference

(Scheiling et al., 2017). Nearby, Herrera et al. (2016) suggest that there may be a small amount of recharge from Laguna Tuyajto in Chile to the underlying Salar de Aguas Calientes-3 basin. Furthermore, they state that, despite differences in buoyancy between saline brines and fresh groundwater, some proportion of the saline water from Laguna Tuyaito likely mixes and becomes incorporated with regional groundwater flow.



In comparison, few hydrogeology studies have examined groundwater recharge zones in southern Peru. The city of Arequipa (2300 m.a.s.l.) is Peru's second largest, situated near a high-elevation (4300 m.a.s.l.) closed-basin salar, Laguna Salinas. Local research conducted on the source of springs east of the city of Arequipa by the Peruvian Agency INGEMMET ("Instituto Geologico, Minero y Metalurgico") indicates that the zone of groundwater recharge is precipitation on a nearby high-elevation volcano called Pichu Pichu (Peña, 2018). However, the potential influence of the Lagunas Salinas basin and the salar itself as

a recharge source for the lower elevation springs has not been thoroughly investigated. Due to the importance of groundwater systems in southern Peru and the hydrogeological complexity of the Central Andes, it is essential to study the hydrogeological behavior of the basins throughout the region. The objectives of this study were to identify sources of recharge near the city of Arequipa using natural geochemical and isotopic tracers in springs, surface waters (rivers and salar), and precipitation (rain and snow). We assessed the contribution of high-elevation precipitation or salar seepage, recharge zones, and groundwater

flowpaths using a combination of stable isotopes ($\delta^{18}O$ and $\delta^2H$), and hydrogeochemical characterization of groundwaters and tritium ($^3H$) to obtain important insights about the zones of groundwater recharge and an estimate of the minimum residence time of groundwater. Specifically, we address the question: is the Laguna Salinas a source of recharge for the regional groundwater system that supports low-elevation springs in the Arequipa region?

## 2 Study area

### 2.1 Arequipa, Peru

Our study is in the Central Andes of southwestern Peru in the department of Arequipa between the Andean Western Cordillera and the coastal belt at 16º20'S latitude and 71º30'W longitude (Figure 1). Local springs and rivers provide drinking and irrigation water for the city. The Río Chili, which flows through the city, and La Bedoya spring to the east are the two main sources of drinking water for the metropolitan area of Arequipa (Sedapar, 2018). Groundwater in Arequipa is also very

important for mining and agriculture. Commercial crop production and self-consumption also rely on these water resources in addition to large-scale water diversion projects from the neighbouring Majes watershed (MINCETUR, 2017; Gerencia Regional de Agricultura de Arequipa, 2015). Water-intensive mining activity occurs throughout Arequipa, with the largest mining company (Cerro Verde) just south of the city (MINCETUR, 2017). Additionally, springs in Arequipa are socio-culturally important and are used for recreation and spiritual purposes (Stensrud, 2019), and have ecological significance.

In arid regions such as Arequipa, the larger dependence on groundwater resources results in increased pressure on this natural resource as population continues to grow. These problems are exacerbated by the effects of climate change as much of the water used in coastal regions is redirected from high elevations where decreased precipitation is projected in the coming century (Neukom et al., 2015). According to census data from 2007 to 2017, population growth in arid coastal areas of Peru had an annual increase of 1.3% which has slowed compared to previous decades (INEI, 2018). Arequipa was the department

with the second highest population growth (1.8% annual increase) and the capital city increased from 806,782 to 1,008,290





people from 2007 to 2017 (2.3% annual increase; INEI, 2018). Continued population growth and a growing agricultural sector will disproportionately affect groundwater resources in southern Peru.

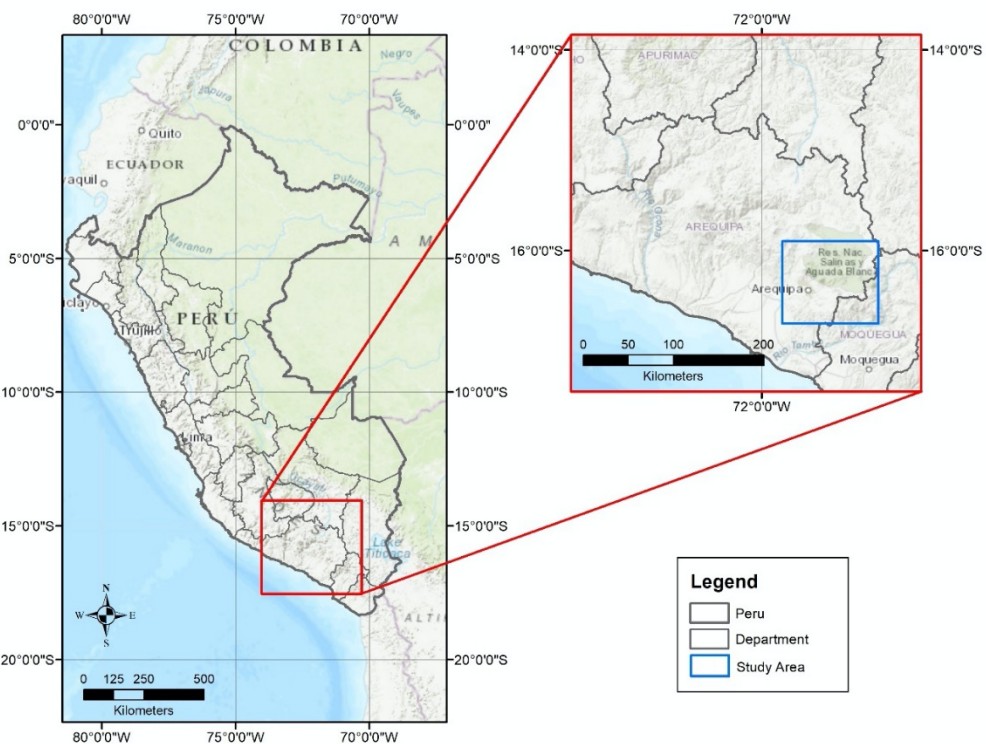

**Figure 1: Map of the Arequipa department in southern Peru, the Quilca-Vitor-Chili rivers basin, and our study area in the**
**northeastern part of this basin.**

## 2.2 Quilca-Vitor-Chili Basin

The study area is located within the Quilca-Vitor-Chili River basin, extending from sea level at the Pacific coast to the Andean Altiplano. Our research was conducted in the slopes of the Western Cordillera in the districts of Characato (~2500 m.a.s.l.),
Chiguata (~3000-3500 m.a.s.l.) and the high-elevation closed-basin salar, Laguna Salinas (~4300 m.a.s.l.), which is approximately 80 km east of the city of Arequipa (Figure 2). The dormant Pichu Pichu volcano (5510 m.a.s.l.) has seasonal snow cover and forms an arcuate ridge between the city of Arequipa and Laguna Salinas. The Laguna Salinas closed-basin salar likely formed during the creation of the late Miocene to early Quaternary volcanic range, which includes the Pichu Pichu, Misti, and Chachani volcanoes (Lebti et al., 2006); and following an eruption of the Pichu Pichu volcano 6.7 Ma (Kaneoka
and Guevara 1984; personal communication with Jean-Claude Thouret, 2020). The volcanic activity in the region cutoff a direct surface-water connection to the ocean and instead, created the internal drainage system of the salar. The Laguna Salinas salar is 67.1 km$^2$ when fully flooded, however the entire closed basin is 142 km$^2$. The active Ubinas Volcano (5672 m.a.s.l.)



is located northeast from Laguna Salinas in a Quaternary volcanic range. While precipitation that falls on Ubinas can drain toward Laguna Salinas, this volcano is geographically close to the Río Tambo, therefore, water falling on Ubinas also drains

to the adjacent Río Tambo basin. We sampled springs emerging within the Río Andamayo watershed draining the southwestern slope of the Pichu Pichu Volcano and with a headcut approaching the Laguna Salinas. Several smaller rivers, the Mollebaya, Socabaya, and Characato, also join the Andamayo, although they head at lower elevations. The Río Andamayo joins the Río Chili in the city of Arequipa forming the Río Vitor which flows into the Río Quilca and eventually the Pacific Ocean.

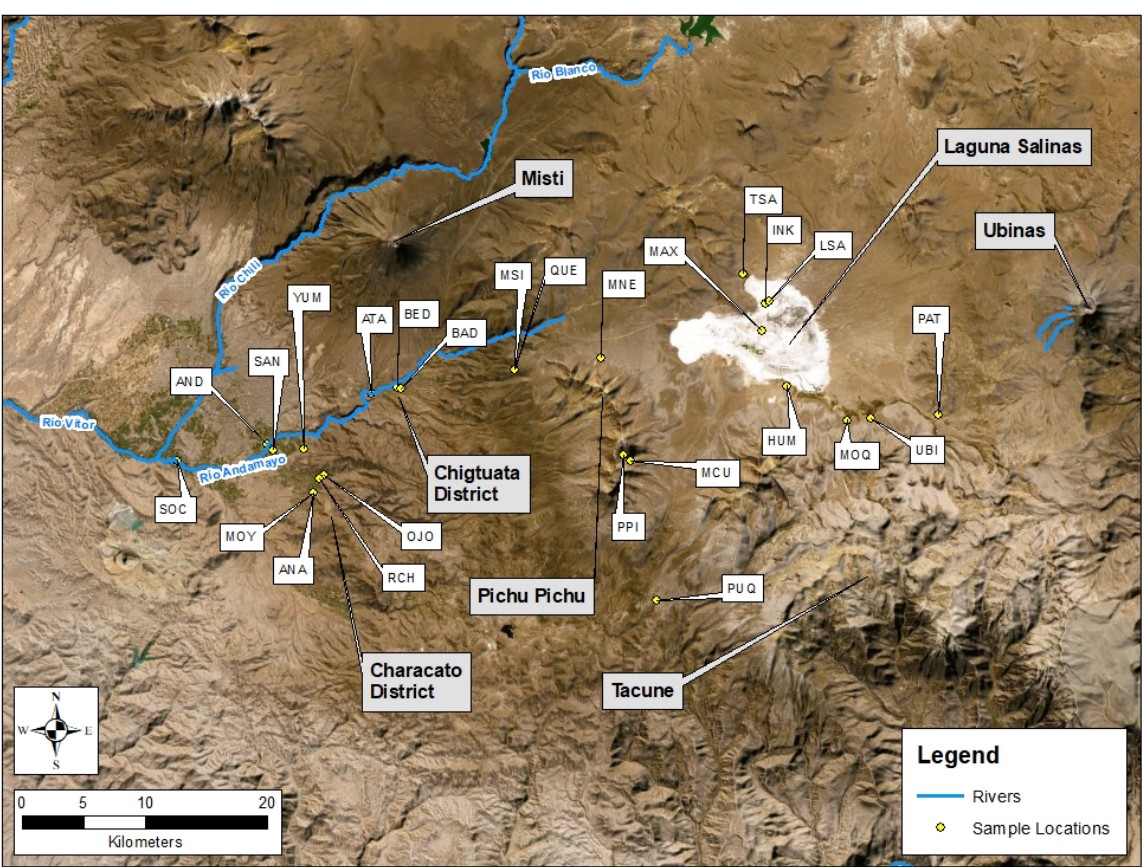


**Figure 2: Map of the study area and sampling locations in yellow dots (credit: Emily E. Frisbee).**



## 2.3 Climate

The western Andes has experienced an arid to semi-arid climate since the modern Andes were uplifted in the Middle Miocene,
forming a barrier to easterly winds (Warren, 2016). By the Pliocene, the southern Peruvian Cordillera was sufficiently uplifted
to produce alpine glaciation, and in the early to middle Pleistocene, several glaciations occurred in this region of the Andes.
However, most Andean glaciers have receded since the mid-19[th] century (Seltzer, 1990). There are currently no glaciers in our
study region (Ehlers et al., 2011).

The western part of the Central Andes today is characterized by a strong annual cycle of precipitation consisting of a very dry
winter (June to August) and a wet summer monsoon season (November to April) bringing precipitation from the Amazon with
large interannual variability (Garreaud, 2009; Urrutia and Vuille, 2009; Imfeld et al., 2020). The Pichu Pichu volcano
commonly accumulates snow during the summer. Average annual precipitation is higher (400-600 mm) at elevations above
4000 m.a.s.l. compared with lower elevations in the department of Arequipa (~100 mm) (Moraes et al., 2019). Annual average
temperatures are cooler at higher elevations and warmer at lower elevations near the coast, except for near large rivers (Moraes
et al., 2019). There are two meteorological stations operated by the Peruvian Servicio Nacional del Meteorolgía e Hidrología
del Perú (SENAMHI) in our study watershed. Those stations are located in the districts of Chiguata and Salinas Huito (near
Laguna Salinas) (Senamhi, 2020). Precipitation data from 2018 showed annual precipitation of 140 mm for Chiguata (2902
m.a.s.l.)  and 350 mm for Salinas Huito (4349 m.a.s.l.). The 2018 annual maximum temperature for Chiguata and Salinas Huito
were 20.3ºC and 12.7ºC, respectively, while the minimum temperatures during this year were 5.9ºC and -4.7ºC. Data for 2019
had many gaps and quality control issues at Salinas Huito so it is not presented here.

Precipitation trends within the department of Arequipa show annual precipitation has increased from 1988 to 2017 most
significantly at the higher elevations during the wet season (Moraes et al., 2019; Moreas et al., 2020). An annual and seasonal
trend analysis of precipitation for the period 1965 to 2018 for the larger Southern Peruvian Andes region showed increasing
precipitation during the summer in most of the stations (Imfeld et al., 2021).

Temperature trends within the department of Arequipa from 1988 to 2017 show increased daily minimum temperatures and
also daily maximum temperatures except in areas of expanding irrigation projects (Moraes et al., 2019; Moreas et al., 2020).
A warming trend of ~0.2ºC increase per decade has been observed from 1981 to 2010 in temperature records from the Central
Andes (Vuille et al., 2015). Other climate studies of the Peruvian Andes have also reported warming trends during the last
decades (Lavado Casimiro et al., 2013; Salzmann et al., 2013) most notably during the spring (Imfeld et al., 2021). While an
overall warming trend is observed for the Central Andes, temperature increases have also been shown to be elevation-
dependent, with greater warming over time occurring at higher elevation than at coastal areas which are more influenced by
ocean temperatures (Viulle et al., 2015; Aguilar-Lome et al., 2019). The warming temperature trends at high elevations play a


role in melting Andean glaciers and decreasing snowpack cover, which are important sources of freshwater during the dry season.

Simulations of future climate indicate precipitation may decrease in the Central Andes. For example, one study estimates a
precipitation decrease of 10%-30% by the end of the 21$^{st}$ century (Minvielle and Garreaud, 2011). Another study similarly determined that precipitation could decrease through the mid-21$^{st}$ century (Neukom et al., 2015). While these projections are uncertain, potential precipitation reduction could be a great concern for the dry region of the Central Andes, since this would cause severe changes to the hydrological cycle and further intensify the regional water vulnerability.

### 2.4 Geologic and hydrologic description

The Chili watershed drainage can be divided in three main geomorphologic landforms largely shaped by tectonic and volcanic processes: 1) the Arequipa depression, 2) the Western Cordillera which comprises the Grupo Barroso and cluster of volcanoes, and 3) the highland hills and depressions surrounding Laguna Salinas (Peña, 2018; Thouret et al., 2001). The geology of the region is complex reflecting the orogenic history of the Andes and thus has bearing on the intricacy of local hydrology, in particular, groundwater flowpaths. The Charcani gneiss, a low permeability Pre-Cambrian feldspathic basement rock of the
Arequipa massif (Jenks, 1948; Peña, 2018), limits hydrologic conductivity at depth and creates a barrier to groundwater flow (Guevara, 1965) and possibly deep circulation.   This rock is exposed in small outcrops along river channels of the districts of Mollebaya and Yarabamba.

Thick volcanic fills and carbonate platform formations, associated with extensional processes during back-arc formation of
the Arequipa-Tarapaca Basin overlie the Charcani gneiss (Sempere et al., 2002; Vicente, 2006). Thick sediments of the Mesozoic Socosani formation with sequences of limestones interbedded with black shales and sandstone are ca. 80 m thick in Arequipa (Jenks, 1948). Detrital clastic sediments of the Grupo Yura contain sequences of sandstone interbedded with thin layers of shales and limonites. The Grupo Yura is overlain by younger volcanic deposits from the late Miocene to Pleistocene from the Grupo Barroso, which is characterized by porphyritic andesite lavas from volcanoes Pichu Pichu and Misti. Local
confined and unconfined aquifers have previously been identified within permeable units interbedded between baked contacts within the Grupo Barroso (Peña, 2018). Fluvioglacial fill deposits on the slopes of Pichu Pichu Volcano lay beneath more recent pyroclastic and clastic materials (Thouret et al., 2001). A landslide of pyroclastic deposits ca. 400 m thick outcrop along the Río Andamayo gorge with a pseudotachylyte basal contact with the underlying ignimbrite may limit deep circulation in the Characato area (Legros et al., 2000). Springs in our Chiguata study site lay above the Chiguata paleolake deposits
comprised of high permeability sandstones and thin units of low permeability stratified diatomite and clay (Garcia et al., 2016). Finally, alluvial and fluvial deposits consisting of mainly gravel, sand, silt, and poorly consolidated conglomerates are distributed between Mollebaya and Arequipa in the banks of the Ríos Chili, Andamayo, and Yarabamba.



Faults are common throughout the study area (Figure 3). Some of the faults are associated with volcanic activity (ring-fracture
faults and normal faults) and others are associated with strike-slip plate motion. These faults are likely hydrogeologically
important because the damage zone associated with the fault (including fault breccias and fractures) can increase permeability
and enhance groundwater flow and/or act as barriers bringing groundwater to the surface. The volcanic complexes (Misti and
Pichu Pichu) along the Western Cordillera are aligned along a strike-slip fault that trends from NW to SE (Mering et al., 1996;
Thouret et al., 2001). While many faults have been mapped in this region (Figure 3, Thouret et al., 2001; Lebti et al., 2006;
Bernard et al., 2019), the most relevant fault to our study are the normal faults (NW to SE direction) present in the district of
Chiguata (Benavente et al., 2017, Thouret et al., 2001). The Río Andamayo is fault bounded and its entire course flows along
a fault separating the older Pichu Pichu mountain block from the uplift associated with Misti (Figure 3).

In the highlands, the salar Laguna Salinas is 14 km from east to west and 8 km from north to south (Garrett, 1998). The lake
floods to an average depth of 50 cm during the rainy season from December to March, and then dries during the long dry
season. Economically important evaporite minerals form in the salar during the dry season. These minerals, ulexite, halite,
Glauber salt, and thenardite, are found at the crust of the salar and with depth in the lacustrine deposits (Muessig, 1958; Garrett,
1998). The lacustrine deposits consist of layers of halite and thenardite ($Na_2SO_4$) at the surface (5-15 cm), followed by a thin
white volcanic ash layer (< 15 cm), various mud layers containing ulexite ($NaCaB_5O_9*8H_2O$; ~ 1.5-1.8 m), and the main
ulexite bed occurs between 0.25 to 1.3 m in the lacustrine sediments. Inyoite ($Ca_2B_6O_{11}*13H_2O$) is also present beneath the
ulexite bed along the eastern side of the salar near the Tusca Hot Spring (<15 cm; Muessig, 1958; Garrett, 1998).

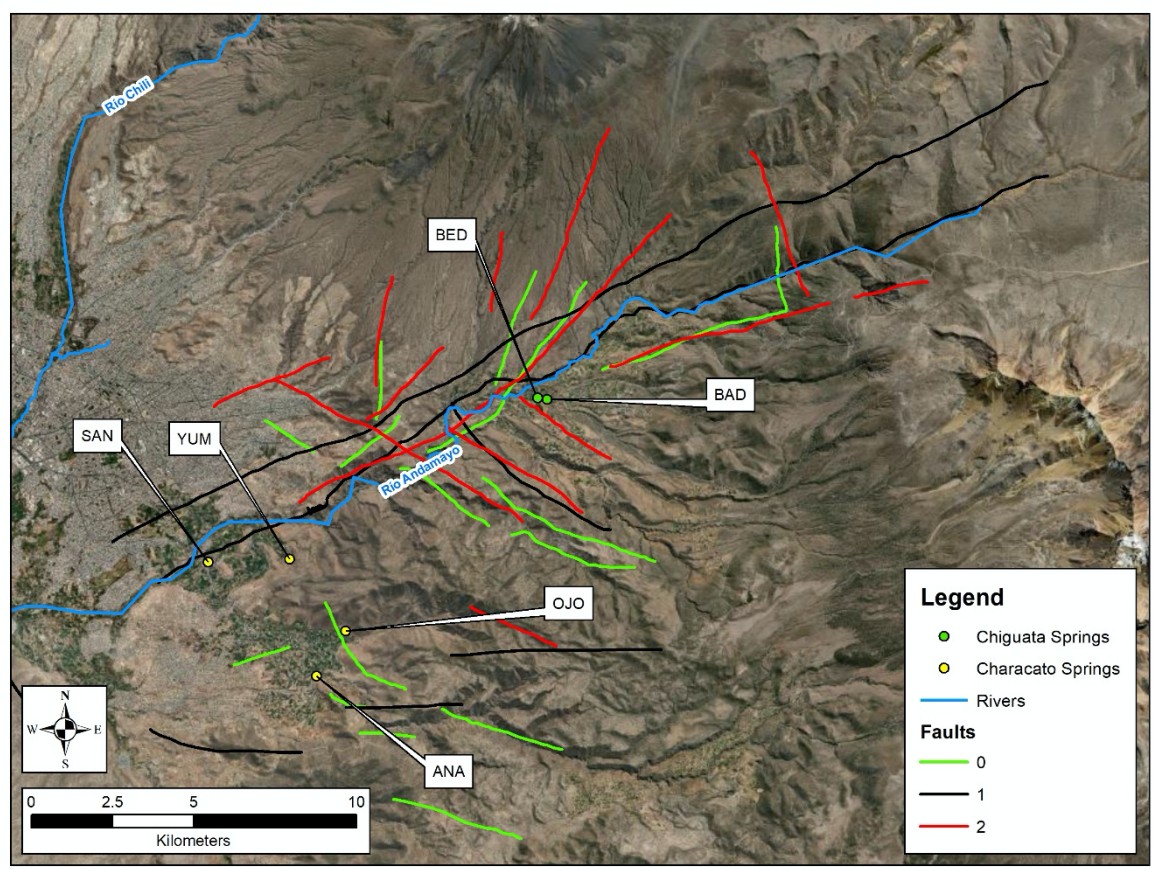

**Figure 3: Faults identified in the study area (credit: Emily E. Frisbee). Green lines are faults identified in Bernard et al., 2019; black lines are faults identified in Lebti et al., 2006; and red lines are faults identified in Thouret et al., 2001. The Río Andamayo is fault-bounded (black lines running parallel to the river). The La Bedoya (BED) and El Badén (BAD) springs are located in the Chiguata District is located near the (BED) and (BAD) springs in green. The springs Sabandia (SAN), Yumina (YUM), Ojo de Chrarcato (OJO), and Santa Ana (ANA) are located in the Characato district.**

## 3 Methods

### 3.1 Water sampling

Table 1 lists coordinates, elevation, names and IDs of water samples collected in this study. We sampled a total of six springs once a month from March 2019 to February 2020. Four springs (Yumina, Sabandía, Santa Ana, and Ojo de Characato) were obtained in the district of Characato, and two springs (La Bedoya and El Badén) were sampled in the district of Chiguata (Figure 2, Table 1). We collected river water samples once every two months at two locations (higher elevation at Chiguata and lower elevation at Characato) on the Río Andamayo, which drains from the highlands to the city of Arequipa, and at three





smaller rivers: Ríos Socabaya, Mollebaya, and Characato. We sampled Laguna Salinas surface water at the end of the rainy

season (April 2019) and during the dry season (October 2019). We sampled smaller six high-elevation springs without formal

names (TSA, INK, HUM, UBI, MOQ, and PAT) once in October 2019 near Laguna Salinas. The spring INK emerges at the

salar, and this spring is only accessible during the dry season. During the wet season, the INK spring is covered by standing

water. The spring TSA emerges near the town of Salinas Huito, while HUM, UBI, MOQ and PAT emerge from the lower

slopes of the Tacune mountain, south of the salar. We sampled snow from the Pichu Pichu volcano once in April and two high-

elevation springs in the area in May 2019. We also sampled two springs in the high elevation queñuales forest (3500-4100

m.a.s.l.), named for the tree species (*Polylepis* sp.) endemic to the high Andean forest ecosystems above 3500 m.a.s.l., in May

and October 2019. The queñuales forest provides habitat for endemic and threatened species of the Andes; however, these

forests are highly threatened due to timber extraction and the indirect effects of grazing (Camel et al., 2019). An Arequipa

Nexus Institute pluviometer installed in Characato (-16.475038, -71.491583) collected rainwater (Welp et al., *in review*).

Water samples were collected into 25-mL and 250-mL plastic bottles using a hand-pump and filtered using a 0.45 μm

membrane in the field for stable isotope and general chemistry analysis respectively. Additional 1-L samples were collected

at 4 sites in September 2019 for tritium ($^3$H) analyses. All plastic bottles were tightly closed and sealed with electrical tape to

prevent water leakage. Water temperature (°C) and pH were measured in-situ at all sites. Field specific conductivity (SpC, μS

cm$^{-1}$) and total dissolved solids (TDS, ppm) were measured when possible with a YSI Professional Plus (Quatro) multi-

parameter probe. (Supplemental Table 1).



**Table 1. Location, sample ID, elevation, and months sampled of springs, snow, and surface waters. These sites are mapped in Figure 2.**

| Sample Group/Location | Sample ID/Name | Sampling Dates | Elevation (m.a.s.l.) | Lat | Lon |
|---|---|---|---|---|---|
| **Characato Springs** | SAN (Sabandia) |  | 2431 | -16.4497 | -71.4955 |
|  | YUM (Yumina) |  | 2590 | -16.4488 | -71.4730 |
|  | ANA (Santa Ana) | Once a month from 03/2019 to 02/2020 | 2518 | -16.4812 | -71.4655 |
|  | OJO (Ojo de Characato) |  | 2595 | -16.4687 | -71.4575 |
| **Chiguata Springs** | BED (La Bedoya) |  | 2899 | -16.4042 | -71.4044 |
|  | BAD (El Baden) |  | 2900 | -16.4046 | -71.4018 |
| **Laguna Salinas Springs** | TSA (Tambo de Sal) |  | 4332 | -16.3204 | -71.1507 |
|  | HUM |  | 4330 | -16.4029 | -71.1182 |
|  | INK | 10/2019 | 4324 | -16.3423 | -71.1342 |
|  | MOQ |  | 4346 | -16.4286 | -71.0742 |
|  | PAT |  | 4429 | -16.4241 | -71.0072 |
|  | UBI |  | 4347 | -16.4269 | -71.0567 |
| **Pichu Pichu Springs** | MNE | 05/2019 | 4657 | -16.3823 | -71.2549 |
|  | MCU |  | 4888 | -16.4577 | -71.2326 |
| **Queñuales Springs** | MSI | 05/2019 | 4052 | -16.3908 | -71.3183 |
|  | PUQ | 10/2019 | 3770 | -16.5605 | -71.2141 |
|  | QUE | 10/2019 | 4011 | -16.3908 | -71.3184 |
| **Volcano Pichu Pichu Snow** | PPI | 05/2019 | 5106 | -16.4537 | -71.2385 |
| **Laguna Salinas Surface Water** | LSA | 04/2019 and 10/2019 | 4317 | -16.3405 | -71.1313 |
|  | MAX | 10/2019 | 4322 | -16.3624 | -71.1361 |
| **Characato Rivers** | SOC (Socabaya) | 03, 06, 09/2019 and 02/2020 | 2206 | -16.4581 | -71.5656 |
|  | AND (Andamayo in Characato district) | 03, 04, 09/2019 | 2419 | -16.446 | -71.4997 |
|  | MOL (Mollebaya during wet season) | 03/2019 | 2401 | -16.4893 | -71.5020 |
|  | MOY (Mollebaya starting the dry season) | 06, 09, 12/2019 and 02/2020 | 2516 | -16.4815 | -71.4656 |
|  | RCH (Characato) | 03, 04, 06, 09, 12/2019 and 02/2020 | 2541 | -16.4713 | -71.4618 |
| **Chiguata River** | ATA (Andamayo in Chiguata district) | 03, 04, 06, 09, 12/ 2019 and 02/2020 | 2783 | -16.4082 | -71.4234 |
| **Characato precipitation** | N/A | Daily in 2019 | 2300 | -16.4750 | -71.4916 |



### 3.2 Isotopic analysis

Stable isotopes of water ($^2$H and $^{18}$O) were measured at Purdue University on a Liquid Water Isotope Analyzer (Los Gatos Research, San Jose, California USA, model T-LWIA-45-EP). The first 4 injections of 10 total were excluded due to memory effects and the last six injections were averaged for the reported value. Data is reported relative to the international VSMOW-SLAP standard scale (Vienna Standard Mean Ocean Water-Standard Light Antarctic Precipitation), with an analytical error of <0.2‰ and <1.0‰ for $\delta^{18}$O and $\delta^2$H, respectively. The weighted average isotopic composition of rain in Characato at the La

Pampilla Senamhi Station and the local meteoric water line (LMWL, $\delta^2$H = 8.12 $\delta^{18}$O + 10.57‰) within the city of Arequipa was obtained from daily sample collections performed by the Arequipa Nexus Institute from January to March 2019 (Welp et al., in preparation).

Tritium ($^3$H) was analyzed to identify the mixing relationships, sources of recharge, and the presence/absence of modern

recharge. Snow from the Pichu Pichu Volcano summit (PPI), rain from the city of Arequipa, two high-elevation springs near Laguna Salinas, one spring from Characato (OJO) and another spring from Chiguata (BED) were analyzed for $^3$H at the University of Miami, Tritium Lab. The reported accuracy and precision for ultralow activity electrolytic enrichment is 0.1 TU.

### 3.3 Chemical analysis

Cation (Ca$^{2+}$, Mg$^{2+}$, Na$^+$, and K$^+$) and anion (Cl$^-$, Br$^-$, SO$_4^{2-}$, and HCO$_3^-$) analyses were conducted by the New Mexico Bureau

of Geology & Mineral Resources. Cations were measured by inductively coupled plasma optical emission spectrometry (ICP-OES, PerkinElmer Optima 5300 DV) following EPA Method 200.7 (USEPA, 1994). Anions were measured by ion chromatography (IC-Dionex ICS-5000) following EPA Method 300.0 (USEPA, 1993). Boron concentrations were determined using ICP-OES at Purdue University (iCAP 7400, Thermo Scientific, China) with dual plasma view, equipped with Qtegra ISDS software. Boron concentration standards ranged from 2 to 2,000 ppb and had a correlation coefficient of 0.9959.

## 4. Results

### 4.1 Isotopic composition and time series of springs, precipitation, and salar water

The monthly isotopic composition of Characato and Chiguata springs was largely invariable over time (Table 2). The stable isotopic composition of the springs in the district of Characato ranged from -8.7 to -10.1‰ for $\delta^{18}$O and -58.2 to -67.8‰ for $\delta^2$H (Figure 4a, b, c). Springs in the district of Chiguata have a more depleted isotopic composition (-11.6 to -12.9‰ for $\delta^{18}$O

and -85.0 to -97.2‰ for $\delta^2$H), and snow from the Pichu Pichu Volcano and springs around Laguna Salinas area are even more depleted (-14.1 to -16.7‰ for $\delta^{18}$O and -109.4 to -126‰ for $\delta^2$H). Spring samples collected at the queñuales forest zone have an isotopic composition similar to that found in the Characato district, and both have values similar to the weighted average of rain sampled in Characato ($\delta^{18}$O = -9.5 +/- 0.71‰, $\delta^2$H = -65.7 +/- 5‰) (Figure 5a, b, c). Note that high elevation





precipitation other than Pichu Pichu snow was not sampled in this study. Samples from Laguna Salinas surface water that were

obtained in the dry season (October 2019) had a highly enriched isotopic composition, while Laguna Salinas surface water sampled during the rainy season (April 2019) had an isotopic composition of -16.3‰ $\delta^{18}$O and -122‰ $\delta^{2}$H, similar to springs in that area (Figure 5a).

The values of $^{3}$H analyzed in modern rainfall from the city of Arequipa and in snow from Pichu Pichu summit were 1.4 TU

and 2.2 TU, respectively. In contrast, the tritium concentration of the springs in the Lagunas Salinas basin, INK and UBI, were effectively tritium dead (0 ± 0.09 TU). OJO in Characato (0 ± 0.04 TU) and the spring BED in Chiguata (0.12 ± 0.04 TU) were also considered tritium dead since the detection limit for low-tritium analyses is 0.1 TU.

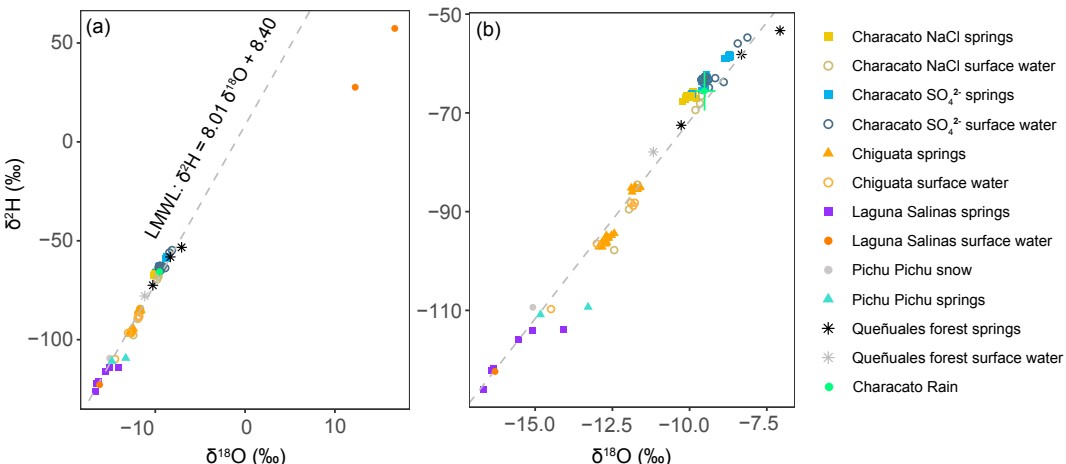

**Figure 4: a) Isotopic composition of springs, rivers, snow, and Laguna Salinas surface water sampled in our study area with the local meteoric water line (LMWL, dashed grey line) estimated for this region. b) Same as (a) after removing dry season Laguna Salinas surface water. c) Same as (b) after removing Characato, Chiguata, and Queñuales forest surface waters. Sample classification (NaCl and SO$_4^{2-}$) based on chemistry data presented in section 4.2.**






**Table 2.** Average isotopic composition for springs sampled from March 2019 to February 2020 and comparison with isotope values of three springs sampled in 2009 by Sulca et al. (2010). Parentheses are standard deviations of all measurements at that site. Discharge from Diaz et al. (1978) compared with measurements in 2016 from Peña Laureano (2018).

| Spring Name | This study March 2019 to Feb 2020 | | Sulca et al. (2010) Sampled Nov 2009 | | Peña Laureano (2018) Sampled 2016 | Diaz et al. (1978) Sampled 1977 |
|---|---|---|---|---|---|---|
| | $\delta^{18}O$ (‰) | $\delta^{2}H$ (‰) | $\delta^{18}O$ (‰) | $\delta^{2}H$ (‰) | Discharge (L/s) | Discharge (L/s) |
| Yumina | -10.1 (± 0.1) | -66.9 (± 0.4) | -10.2 | -66.3 | 215 | 255 |
| Sabandia | -9.9 (± 0.1) | -66.1 (± 0.4) | | | | |
| Santa Ana | -9.1 (± 0.6) | -61.1 (± 3.6) | | | | |
| Ojo de Characato | -9.5 (± 0.1) | -63.0 (± 0.6) | -9.7 | -63.1 | 206 | 216 |
| La Bedoya | -11.7 (± 0.1) | -85.3 (± 0.3) | -11.8 | -84.8 | 114 | 206 |
| El Baden | -12.7 (± 0.2) | -95.9 (± 1.0) | | | | |


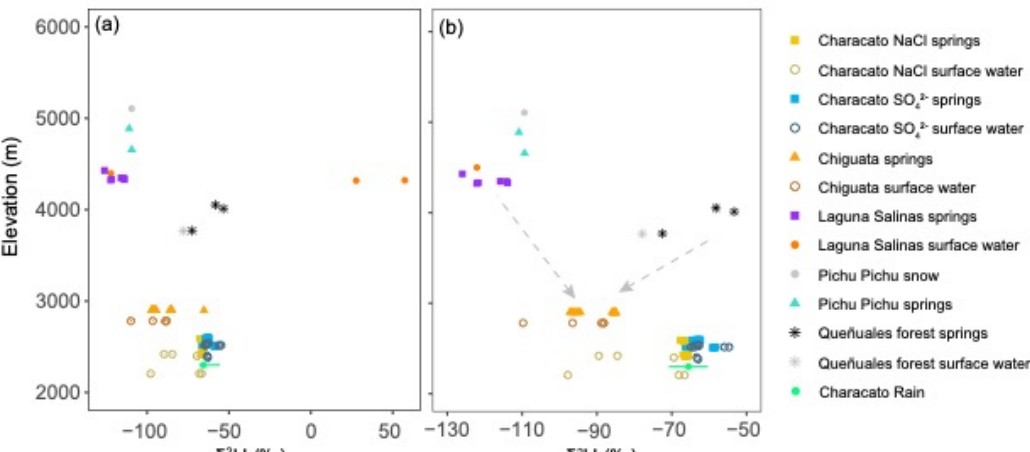

**Figure 5: a)** Deuterium composition with respect to elevation of springs, rivers, and Laguna Salinas surface water from high-elevation Laguna Salinas to lower elevations in the district of Characato. Snow from Pichu Pichu Volcano and weighted average of
rain in Characato are also shown with standard deviation. **b)** Same as (a) after removing the isotopically enriched dry-season Laguna Salinas surface water. **c)** Same as (b) after removing Characato, Chiguata, and Queñuales forest surface waters. Dashed arrows indicate hypothesized mixing lines between high-elevation sources of recharge. Sample classification (NaCl and $SO_4^{2-}$) based on chemistry data presented in section 4.2. Lagunas Salinas surface waters varying dramatically from positive $\delta^{2}H$ values during the dry season to -115‰ in the wet season.






## 4.2 Isotopic composition and time series of surface waters

In contrast to nearly isotopically-constant springs, most rivers were more variable (Figure 6). The exception is the Río Characato (RCH), which only varied by 1‰ in $\delta^2$H during our sampling dates. The Río Socabaya (SOC) and the Río Andamayo sampled at Chiguata (ATA) showed more depleted isotopic values in February 2020. The isotopic composition of the Río

Mollebaya (MOL and MOY) was more enriched than most rivers except the Río Characato, and also showed slightly more enriched isotopic composition during June and September compared to other months. The isotopic values of the Río Andamayo sampling site at Chiguata are comparable to the composition of springs sampled in Chiguata (BED and BAD). Likewise, the isotopic composition of samples from the Ríos Mollebaya and Characato are similar to springs sampled in Characato (ANA and OJO), respectively, which are the springs nearest to each of these rivers. These similarities suggest that groundwater

discharge is contributing to river flow in these areas.

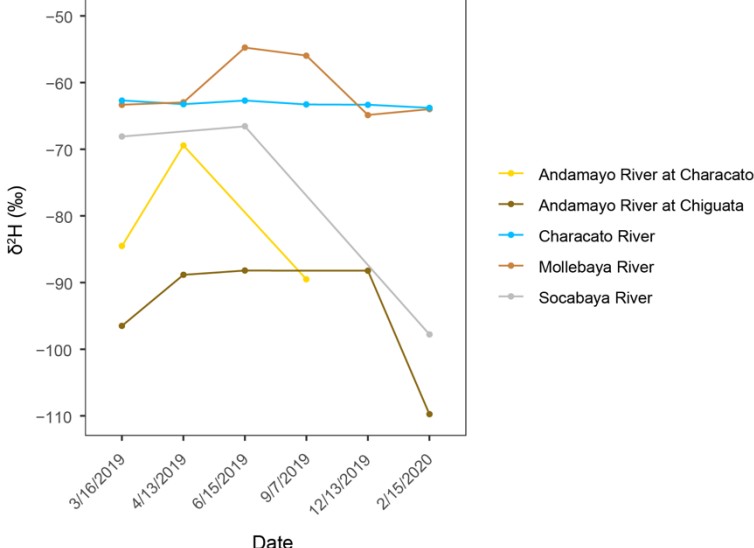

**Figure 6: Deuterium composition time series of rivers in the Characato and Chiguata districts show isotopic compositions similar to nearby springs indicating strong groundwater influence on surface waters.**




### 4.3 Hydrochemical facies and processes

Most chemistry of springs in our study region fall on a mixing line between sodium chloride (NaCl) and calcium sulfate ($CaSO_4$) water types (Figure 7). NaCl type waters include Laguna Salinas surface water, three springs near Laguna Salinas (UBI, MOQ, and INK), and springs in the district of Chiguata (BED and BAD). Pichu Pichu snow, one Pichu Pichu spring
(MCU) and two Characato springs (OJO and ANA) plot towards the $CaSO_4$ quadrant, while the Characato springs that are geographically closer to the Río Andamayo (SAN and YUM) plot in the middle of this inferred mixing line. Since two springs in Characato had greater NaCl percentage (SAN and YUM) than the other two springs (OJO and ANA), we classified samples from Characato into two groups: NaCl and $SO_4^{2-}$ springs.

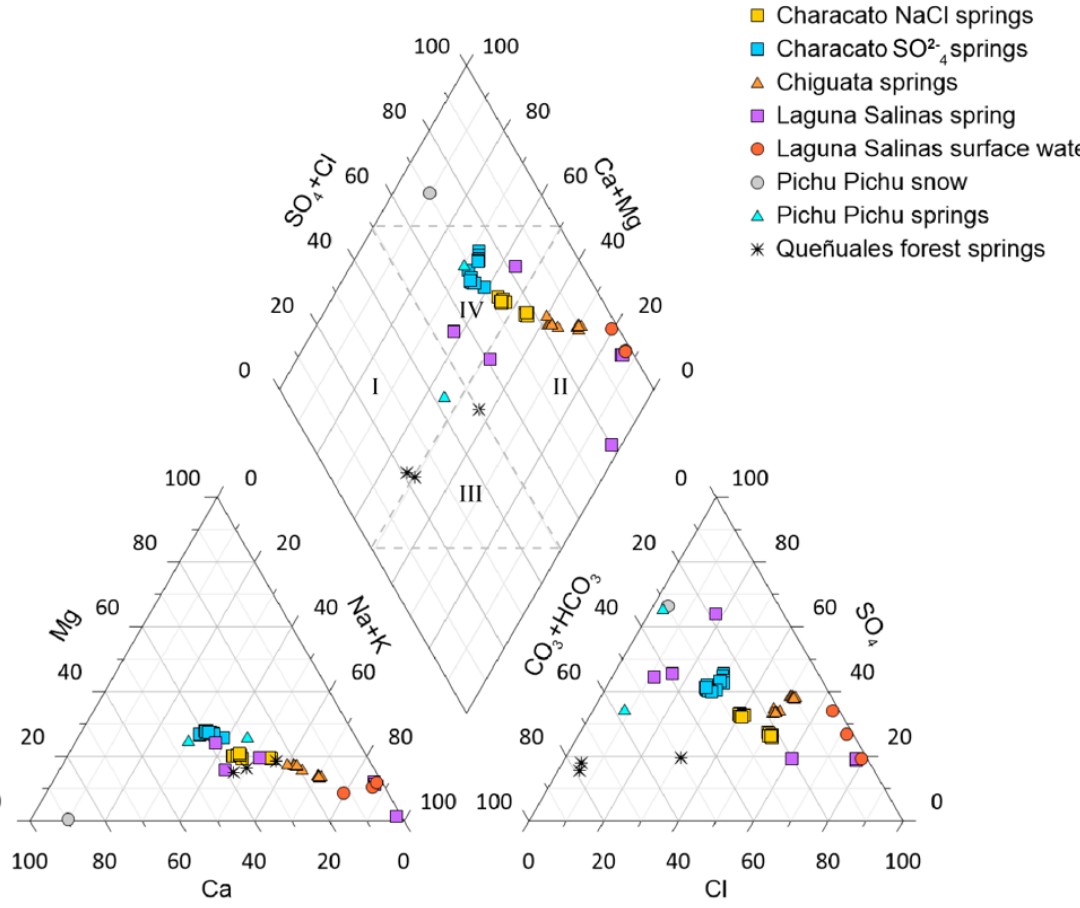


**Figure 7: Piper diagram of all groundwater samples, Lagunas Salinas surface water, and snow sample in this study. Quadrant I represents calcium-bicarbonate waters, quadrant II represents sodium-chloride waters, quadrant III represents sodium-bicarbonate waters, and quadrant IV represents calcium-sulfate waters.**



Gibbs diagrams are commonly used to identify where water samples plot relative to precipitation dominance (rainfall), evaporation dominance, and rock-weathering processes. Marandi and Shand (2018) caution that Gibbs diagrams are not always sufficient when interpreting the processes that affect and control the geochemistry of groundwater. However, the Gibbs diagrams appear robust for springs in Characato and Chiguata and for springs and surface waters near Laguna Salinas. The Gibbs diagrams for the major ion compositions of our springs indicate that water chemistry of Characato and Chiguata springs

is controlled by water-rock interactions (Figure 8a, b). Chiguata springs plot at the distal end of the rock dominance region and closer to the evaporation dominance region than the Characato springs. Two springs near Laguna Salinas (UBI and MOQ) and Laguna Salinas surface water samples fall in the evaporation dominance area, which suggests that their chemistry is influenced by evaporation. Two springs from the queñuales forest and one spring from Pichu Pichu plot towards the precipitation dominance area.


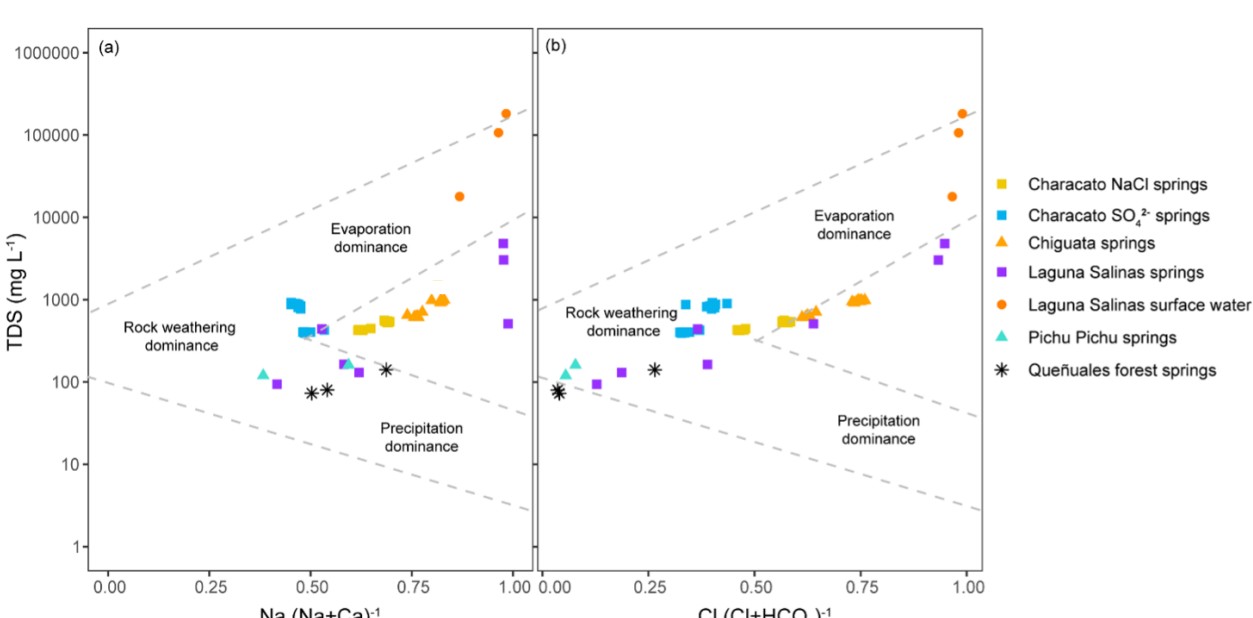

**Figure 8. Gibbs a) cation and b) anion diagram for the ratios of major ion compositions calculated using ppm units. Springs in our study area show a distribution of evaporation and rock weathering influence on the groundwater chemistry. High elevation springs on Pichu Pichu and the queñuales forest are most strongly influenced by precipitation.**






High-elevation springs sampled at Pichu Pichu and the queñuales forest area had low Cl⁻ concentrations (Figure 9). However, high-elevation springs sampled around Laguna Salinas had a wide range of concentrations (Figure 9a). The Laguna Salinas springs UBI and MOQ had high Cl⁻ concentrations of 2150 mg L⁻¹ and 1400 mg L⁻¹ respectively (Figure 9a); while springs

INK, HUM, TSA, and PAT had Cl⁻ concentrations of 154, 40, 6.5, and 4 mg L⁻¹, respectively (Figure 9b). Overall, springs in Chiguata had greater Cl⁻ concentrations than springs in Characato, which are at a lower elevation. This suggests that the springs are not on a single geochemical evolutionary pathway from high elevation to low elevation but are instead supported by groundwater flowpaths from different areas of the watershed.

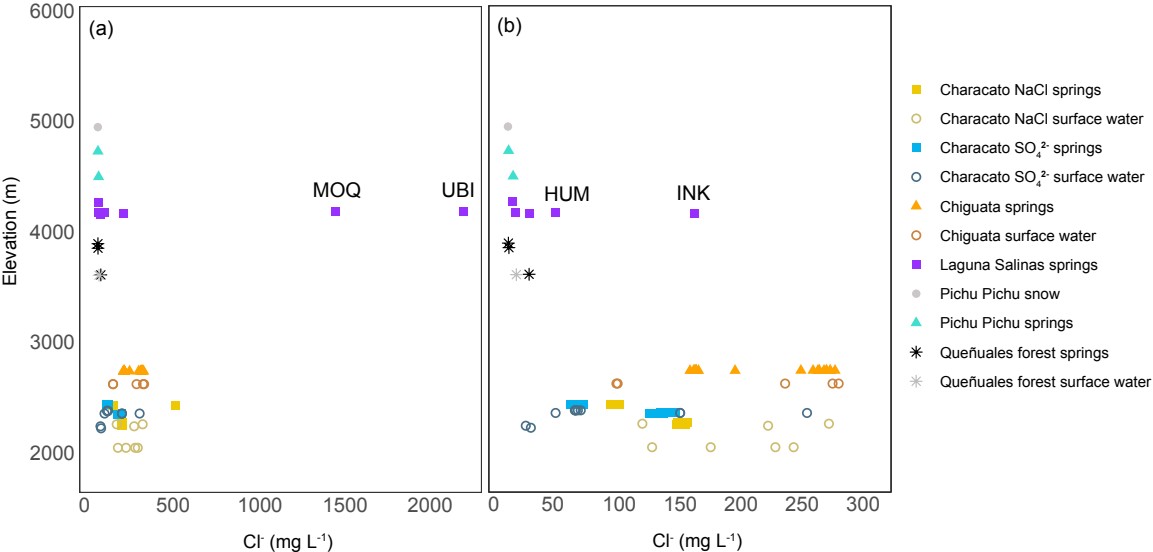


**Figure 9. a) Chloride concentration of all water samples at different elevations, excluding Laguna Salinas surface water, and b) excluding two Laguna Salinas springs (UBI and MOQ) with high Cl⁻.**

The relationship between Cl⁻ and SO₄²⁻ shows two trends: one for Chiguata (BED and BAD) and Characato NaCl (YUM and

SAN) springs with higher Cl⁻ located along the Río Andamayo fault system, and another one for Characato SO₄²⁻ springs with lower Cl⁻ located in a secondary cluster of shorter faults (Figures 3 and 10). This relationship between Cl⁻ and SO₄²⁻ highlights mixing between recharge from Laguna Salinas (high in Cl⁻ and SO₄²⁻) and MBR from precipitation occurring on the Pichu Pichu volcano (low in Cl⁻ and SO₄²⁻). Additionally, the mixing lines (Figure 10) with two different slopes indicates that Chiguata and Characato NaCl springs have a Laguna Salinas mixing endmember with higher Cl⁻ than the Characato SO₄²⁻

springs. The range in Cl⁻ from springs sampled around Lagunas Salinas (Figure 9b) indicates that there are multiple groundwater flow paths in the high elevations.

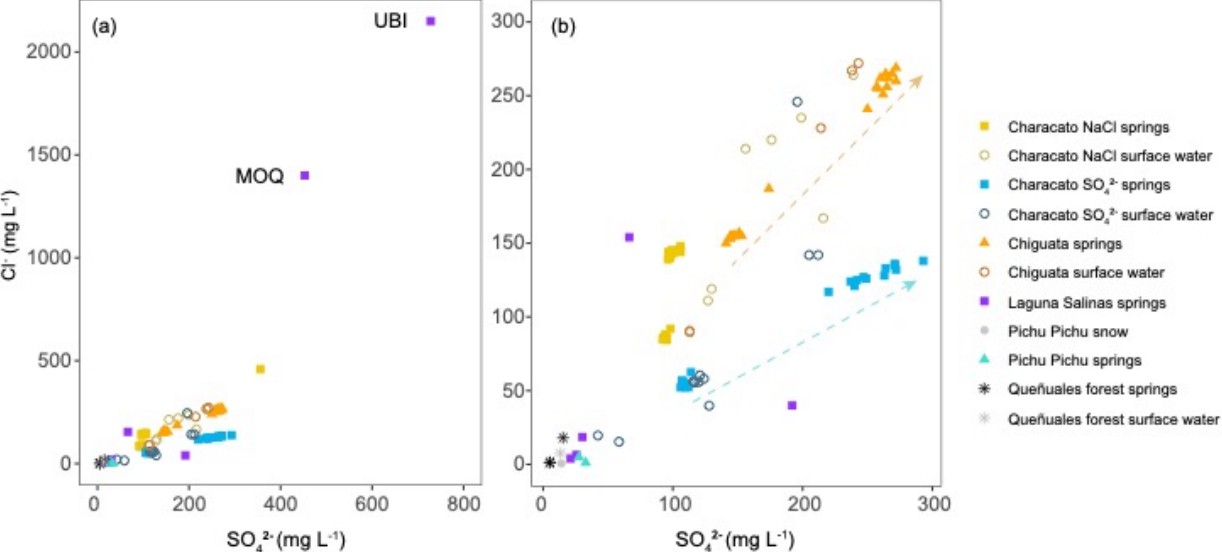

**Figure 10. Relationships between Cl⁻ and SO₄²⁻ of a) all water samples, except Laguna Salinas surface water, and b) excluding two Laguna Salinas springs (UBI and MOQ) with high Cl⁻. Dashed lines indicate patterns of chemical evolution or mixing and not statistical trend lines. The orange evolution line connects springs along the Río Andamayo fault system and the blue line connects springs in the Characto district furthest from the Río Andamayo in a separate cluster of shorter faults.**

## 5 Discussion

### 5.1 Groundwater recharge zones and temporal variability

### 5.1.1 Elevation isotopic lapse rate

In most orographic precipitation studies along mountain ranges, the stable isotopic composition of precipitation becomes progressively more depleted (lighter) at higher elevations (Dansgaard et al 1964; Clark and Fritz et al., 1997; Różański et al., 1993). This relationship with elevation is often reflected in springs as well. Our observations show that high-elevation snow, springs, and unevaporated surface waters are isotopically lighter than lower elevations (Figure 5). However, we observed very little difference in the $\delta^{18}O$ and $\delta^2H$ between springs at an average of 2,533 m in Characato and 3,944 m in the queñuales forest, despite the large elevation difference. Furthermore, amount-weighted rain collected in the Characato district had similar isotopic values as the queñuales forest springs. This suggests very little isotopic lapse rate with an elevation difference of ~1,400 m in our study area. Regionally, the isotopic lapse rate of surface waters in the Western Cordillera is estimated to be -1.0‰ $\delta^{18}O$ per 100 m (Bershaw et al., 2016), which is much larger than observed in this study. The factors controlling the isotopic values of precipitation in the Western Cordillera of the Andes are poorly understood, partly due to lack of data (Valdivielso et al., 2020). Various studies in high-elevation mountain ranges have shown that the relationship between isotopic composition and elevation can be complicated by altitude, orographic characteristics of mountain barriers, the relative influence of various moisture sources, the rain-snow transition elevation, local orographic convection, the recycling of surface





waters, the degree of evaporation in arid conditions, and cloud microphysics which can affect the isotopic composition of local precipitation (Moore et al., 2016; Lechler et al., 2013; Bony et al., 2008;  Numaguti, 1999; Kurita and Yamada, 2008; Hren et

al. , 2009). As observed in Figure 4, the springs sampled at the queñuales forest fall very near to the meteoric water line, which suggests that the enrichment of these springs is not caused by enhanced evaporation in this ecosystem. The convergence of Pacific and Atlantic moisture sources and the orographic effects of the Western and Coastal Cordilleras could contribute to this unusual elevation pattern (Welp et al., *in review*).

### 5.1.2 Spring recharge elevation

The similarities in isotopic composition between local rain, springs in the Characato district, and springs in the queñuales forest area indicate that Characato springs are recharged by rainfall near and above the queñuales forest elevation, low-elevation local rainfall, or a combination of both. Springs from the Chiguata district have isotopic values that fall in between lighter values of Laguna Salinas springs and Pichu Pichu snow, and heavier values of the queñuales forest springs (Figure 5b); which suggests that Chiguata springs receive a mix of water from higher elevations including possible contributions from snowmelt,

medium-elevation queñuales forest springs (presumably fed by rainfall near that elevation), and recharge from the Laguna Salinas.

The springs sampled around Laguna Salinas (~4300 masl) have more depleted $\delta^2$H values than snow and springs sampled at Pichu Pichu, the local peak elevation (~5000 masl), and similar or slightly heavier than the isotopic composition of Laguna

Salinas surface water sampled during the rainy season ($\delta^{18}$O = -16.3‰ and $\delta^2$H = -122.1‰). In addition, INK and UBI springs were tritium dead indicating that these springs are discharging a substantial proportion of groundwater that was recharged several hundred years ago. This implies that springs around Laguna Salinas are supported by recharge from high-elevation precipitation in this region, similar to what flows into the Laguna Salinas basin during the wet season, likely from other mountains surrounding the salar like the Ubinas volcano (5672 m.a.s.l.) and Tacune mountains. Furthermore, the high-

elevation groundwater flow paths are long enough to significantly age the water. A subsurface connection between the Ubinas volcano, located northeast of the salar, and the Laguna Salinas has been proposed by other studies (Cruz et al., 2019; Gonzales et al., 2014). However, this connection, if present, must occur within a very small region since a deep-seated fault is located to the north-northeast of Laguna Salinas and the rock units found along this fault are nearly vertical thereby limiting the possibility of a hydrogeologic connection to the northeast. It is unclear how far this fault extends to the south. Springs can be

seen in Google Earth imagery on the Ubinas side of the fault, but not toward the Laguna Salinas basin. In any case, if there is a hydrogeologic connection from Ubinas to the Laguna Salinas basin, it must occur in a narrow zone immediately east of Ubinas. Stable isotope data of a few springs sampled near Ubinas volcano and springs at high elevations in the vicinity of Tacune mountain in the adjacent Río Tambo basin, show values of -15 to -16.7‰ for $\delta^{18}$O and -122.5 to -130‰ for $\delta^2$H (Cruz et al., 2019). While it is plausible that MBR occurring on the slopes of Ubinas supports groundwater flow into the Laguna

Salinas basin, it's not very probable. However, MBR occurring on Tacune does likely support groundwater flow into the





Laguna Salinas basin; the emergence of high-elevation springs on Tacune cannot be explained otherwise. One other explanation for the depleted isotopic composition of Laguna Salinas springs is paleorecharge from the Last Glacial Maximum (LGM) glacial meltwater that is stored in the basin. We do not have data to test this conceptual model, but glaciers were present in the vicinity of the Laguna Salinas basin during the LGM (Seltzer, 1990; Juvigné et al., 1997). Additional sampling and
dating of springs and wells in the area are necessary to thoroughly test LGM influence.

The long-term stability of the Characato spring isotopic values within our study period and a sampling from a decade ago indicates that they are supported by a stable source of recharge that is nonresponsive to the temporal short-term variability in rain events during the wet season. Three of our springs (YUM, OJO, and BED) were also sampled in 2009 by Sulca et al.
(2010) and the stable isotopic results obtained in their study were nearly identical to ours 10 years later (Table 2). However, a comparison of spring discharge measurements made once in 1977 and once in 2016 reveals that the discharge from some springs has decreased over this 40-year interval. Spring discharge of Yumina, Ojo de Characato and La Bedoya was 255, 216, and 206 L/s, respectively, in 1977 (Diaz, 1978); and 215, 206, and 114 L/s, respectively, in 2016 (Peña Laureano, 2018), reflecting decreases of 16%, 5%, and 45% respectively. Differences in discharge may also be due to the variability in time-of-
year the measurements were taken, changes to the La Bedoya spring channel for water distribution, and possible differences in flow-measurement methodology.

## 5.2 Contribution of groundwater to river flow

While springs showed very little isotopic variability, river isotope data was more variable. This can be expected for surface runoff since streams and rivers integrate many different flowpaths and show an immediate response to precipitation events
during the rainy season causing large variations in surface water isotopic values depending on precipitation isotope values. In comparison, the response of springs to precipitation events is slower since groundwater response times are controlled by aquifer diffusivity (the ratio of transmissivity to storability) and are therefore lagged relative surface-water responses. The upstream reach of the Río Andamayo (ATA) in Chiguata showed a lighter isotopic composition during the rainy season (March 2019 and February 2020), which likely indicates the direct runoff input of high-elevation rain. The Río Socabaya (SOC) shows a
more depleted isotopic value during February 2020 as the Río Andamayo feeds into the Río Socabaya (Figure 6). The similarity in isotopic composition between rivers and nearby springs also suggests that the rivers are either being supported by discharge from springs or direct discharge from groundwater feeding the springs. This is especially true for the Río Characato, which shows very little change in its isotopic values (Figure 6) throughout our sampling period and is extremely similar to its nearest spring, Ojo de Characato (Table 2).

## 5.3 Groundwater residence time

Tritium ($^3$H) in modern rain from Arequipa was 1.4 TU and snow from the Pichu Pichu Volcano was 2.2 TU. These values are comparable to other modern $^3$H values found in this region of the Central Andes (Aravena et al., 1999; IAEA, 2012). For





example, rainwater collected between 1984 and 1986 in northern Chile was reported to have 3 to 10 TU (Aravena et al., 1999),
and precipitation sampled between 2006 and 2012 in Lima, Peru had $^3$H values that ranged between 2.5 and 5 TU (IAEA,
2012). The atmospheric breakthrough of $^3$H associated with nuclear weapons testing in the 1950s and 1960s for South America
peaked at approximately 60 TU in 1965 (Albero & Panarello, 1981). The atmospheric breakthrough of $^3$H was much higher in
the northern hemisphere, for example Miami peaked at 384 TU in 1964. In principle, "young aquifers" containing bomb-pulse
recharged waters in South America should have a $^3$H activity of about 2.9 TU assuming well-mixed conditions. For context,
Moran et al. (2019) show evidence for "fossil water" defined as groundwater older than 60 years (essentially pre-bomb
recharge) in the Salar de Atacama in Chile. In comparison, all of the springs analyzed for tritium in the Arequipa region,
including two at high elevations in Laguna Salinas basin (INK and UBI) and two at lower elevations (OJO and BED), are $^3$H
dead, indicating that these springs are discharging a substantial proportion of groundwater that was recharged a few hundred
years ago. While it is not possible to provide an accurate residence time based solely on $^3$H data for these springs, we infer that
the residence times of springs in the study area are much greater than 60 years. Simple recharge-weighted mixing models for
a well-mixed aquifer indicate that it may take at least 300 years to become tritium dead (Gleason et al. 2020).  However, we
cannot confirm or dismiss the possibility of paleo-recharge from the Last Glacial Maximum with only tritium estimates.

**5.4 Groundwater flowpaths**

The chemical resemblance between Characato springs and high-elevation Pichu Pichu snow and springs, as well as the isotopic
similarities between Characato springs and springs in the mid-elevation queñuales forest indicate that the Characato springs
are supported primarily by MBR occurring in the mid- to high-elevations of Pichu Pichu. The sample of Pichu Pichu snow had
14.2 mg L$^{-1}$ of SO$_4^{2-}$, which is higher than expected for typical snow.  Higher SO$_4^{2-}$ concentrations in Pichu Pichu snow is
likely due to aeolian deposition of SO$_4^{2-}$ from recent nearby volcanic eruptions like Ubinas. Studies in northern Chile have
shown that volcanic eruptions can produce large amounts of air-fall tephra that is deposited in the surrounding mountains and
endorheic basins (Warren, 2016). Such events have the potential to elevate sulfate concentrations of waters and snow. We
observed a high chemical variability in springs surrounding Laguna Salinas that may be influenced by the mineralogy of the
Tacune mountain block. Unfortunately, there is little-to-no information on the mineralogy of the rocks of Tacune.
The commonly observed trend of Cl$^-$ increasing with decreasing elevation (*i.e.,* increasing flowpath length) was not observed
in this study area. Instead, Chiguata springs have higher Cl$^-$ concentrations even though they emerge at a higher elevation than
springs at Characato. Jenks & Goldich (1956) state Cl$^-$ rich groundwater in the region may be explained by weathering of halite
crystals found in volcanic deposits, namely a salmon-colored sillar, one of several rhyolitic tuffs present in the Arequipa region.
However, the salmon-colored sillar has not been mapped in the Chiguata District to our knowledge. An alternate explanation
for this anomaly is that there is a stronger groundwater connection between the high-elevation salar (a source of Cl$^-$ enriched
water) and Chiguata springs (Figure 9 and 10). However, both the chemistry and isotopic composition of Chiguata springs
point towards a mix of recharge from Laguna Salinas and MBR occurring on Pichu Pichu and potentially other surrounding
peaks including Tacune.





We approximate the geochemistry of the groundwater that flows into the Laguna Salinas basin using the chemistry of the INK spring sampled at the salar during the dry season. The validity of using INK as an endmember for groundwater outflow from the Laguna Salinas basin is assessed here by critiquing the chemistry of INK against the geochemical processes responsible for the formation of the evaporite minerals in the salar. As stated in the Site Description, the type of evaporite deposits changes

with depth in the salar. INK has high levels (2.13 mg L$^{-1}$) of boron (B) which can be explained by B-rich water that enters the salar and subsequent evapoconcentration within the salar during the dry season. Springs HUM and UBI discharging into the basin have high B (greater than 15 mg L$^{-1}$) but outside of the analytical calibration used. When the salar begins to evaporate in the dry season, gypsum (CaSO$_4$*2H$_2$O) and anhydrite (CaSO$_4$) precipitate first, followed by halite (NaCl) and thenardite (Na$_2$SO$_4$) (following the sequence proposed by Garrett, 1998). This increases the concentration of B in the remaining water,

some of which likely infiltrates the lacustrine sediments. The occurrence of ulexite minerals at depths greater than 1.2 m (Garrett, 1998) below the lake surface suggests that the salar level has changed through time and consequently, the vertical movement of infiltrating water today slows once it reaches the black and green muddy clay layers at 1.2 m. Here, the B-rich water accumulates and ulexite begins to precipitate (Garrett, 1998; Muessig, 1958). Garrett (1998) states that the most plausible model for the formation of ulexite nodules rather than crystal formation requires the mixing of upwelling cation-rich

groundwater with downwelling (descending) B-rich water from the surface; the subsequent competition between cation-sorption and mineral precipitation in the clay layers favors nodule growth. Since INK discharges into the basin and is flooded during the wet season, this suggests that the salar serves as a local topographic low which would tend to enhance groundwater discharge or upwelling. Muessig (1958) inferred that the inyoite underlying the ulexite formed in place as a primary mineral from the influx of B-rich water from the nearby hot spring. This specific area of the Laguna Salinas may be similar to the

transition zones in the Salar de Atacama described in Boutt et al. (2016), Moran et al. (2019), and Munk et al. (2021). The evaporation rate during the dry season, 141.3 cm (ANA, 2013) is quite high; however, the influx of surface water to the salar during the wet season is temporarily larger than evaporation rates causing the salar water level to rise seasonally. Therefore, it's plausible that water can leak from the salar during the wet season. In any case, the geochemical composition of the spring INK appears to be a mixture of waters in contact with evaporites and waters whose geochemical composition can be explained

by rock-water interactions (*i.e.,* groundwater flowing from the surrounding mountains and discharging into or around the salar). We conclude that it serves as a reasonable mixed salar basin groundwater endmember.

We propose that groundwater recharged on the neighbouring peaks including Pichu Pichu and Tacune (and potentially Ubinas) flows toward the Lagunas Salinas basin which is a local topographic low. The pre-modern mountain-block groundwater then

mixes with Cl$^-$ rich water that is recharging beneath the salar during the wet season. The resulting Cl$^-$ rich groundwater then flows out of the basin facilitated by extensive faulting in the headwater of the Río Andamayo (Thouret et al. 2001, Bernard et al., 2019; Lebti et al., 2006). The groundwater discharges along a fault at the Chiguata springs. The Río Andamayo is fault-bounded from Laguna Salinas to the city of Arequipa, providing the hydraulic connection between Laguna Salinas and the Chiguata and NaCl Characato springs at lower elevations (Figure 3).






In comparison, the groundwater which supports the Characato $SO_4^{-2}$ springs is geochemically distinct from the Chiguata and Characto NaCl springs. We infer that MBR occurring on Pichu Pichu and the queñuales forest zone supports groundwater flowpaths on the southwestern slopes of the mountain toward the city as well as flowpaths to the northeast into the Laguna Salinas basin. Groundwater flows through distinct sulfate-bearing volcanic deposits toward the Characato District. The

Characato springs emerge along several faults on the lower slopes of Pichu Pichu (Figure 3). It's unlikely that there is significant groundwater circulation on the southwestern slope of Pichu Pichu because this area has been mapped with a landslide deposit overlying a pseudotachylite (Legros et al., 2000) that probably limits deep circulation.

**5.5 Conceptual hydrologic models of high-elevation salar connectivity to low-elevation springs**

We identify two main recharge zones for the lower-elevation Characato and Chiguata springs of our study region based on

isotopic and geochemical analyses: 1) MBR within and above queñuales forest at ~4000 m.a.s.l. and 2) recharge from the Laguna Salinas basin (Figure 11). We also identify two main groundwater flow pathways: 1) inter-basin groundwater flow (IGF) from the Laguna Salinas basin that provides the characteristic higher Cl⁻ concentration to Chiguata and Characto springs along the Andamayo fault system (Figure 11a), and 2) groundwater flowpaths from the Pichu Pichu Volcano to lower elevation springs in Characato and Chiguata (Figure 11b).


Our results indicate substantial MBR within or above the queñuales forest, however, forest recharge zones are controversial. Paired forest-logged catchments studies have often found reduced watershed discharge in forested catchments due to higher evapotranspiration rates (see review by Brown et al., 2005). Observations and ecohydrology modelling of the relationship between canopy cover and MBR indicate that snowpack shielding and decreased sublimation may increase recharge in

mountain forests (Magruder et al., 2009; Biederman et al., 2015). In general, the relationship between vegetation, water yield and MBR remains uncertain (Markovich et al., 2019), even as differences in tree types in a single study region have been shown to change water yield (LaMalfa and Ryle, 2008). Therefore, future studies should investigate the ecohydrology of the queñuales forests and their role in MBR.

There have been several proposed hydrological conceptual models for other similar high-elevation closed-basin salars and their influence on lower elevation springs in northern Chile. Some prior studies suggest that there is some degree of hydrogeologic connectivity between high elevation salars and lower elevation groundwater (Magaritz et al., 1990; Risacher et al., 2003; Herrera et al., 2016; Jayne et al., 2016); however, other studies reject this conceptual model (Uribe et al. 2015, Scheiling et al, 2017). Our conceptual model conforms with the former models, for example, the Herrera et al. (2016) model

in the Laguna Tuyajto of northern Chile, suggesting that closed-basin salars are not simple evaporation pans but instead leakage recharges groundwater flowpaths that support lower elevation springs. Our conceptual model is in partial agreement with the findings from local research that identified the zone of groundwater recharge at the Pichu Pichu Volcano (Peña, 2018).





However, our chemical and isotopic data, as well as inference from fault connectivity, identifies the high-elevation Lagunas Salinas closed-basin salar as an additional contributor to groundwater recharge for lower elevations springs and suggests that

inter-basin groundwater flow is an important hydrogeological process and a major influence on the yearlong flow and chemistry of these springs.

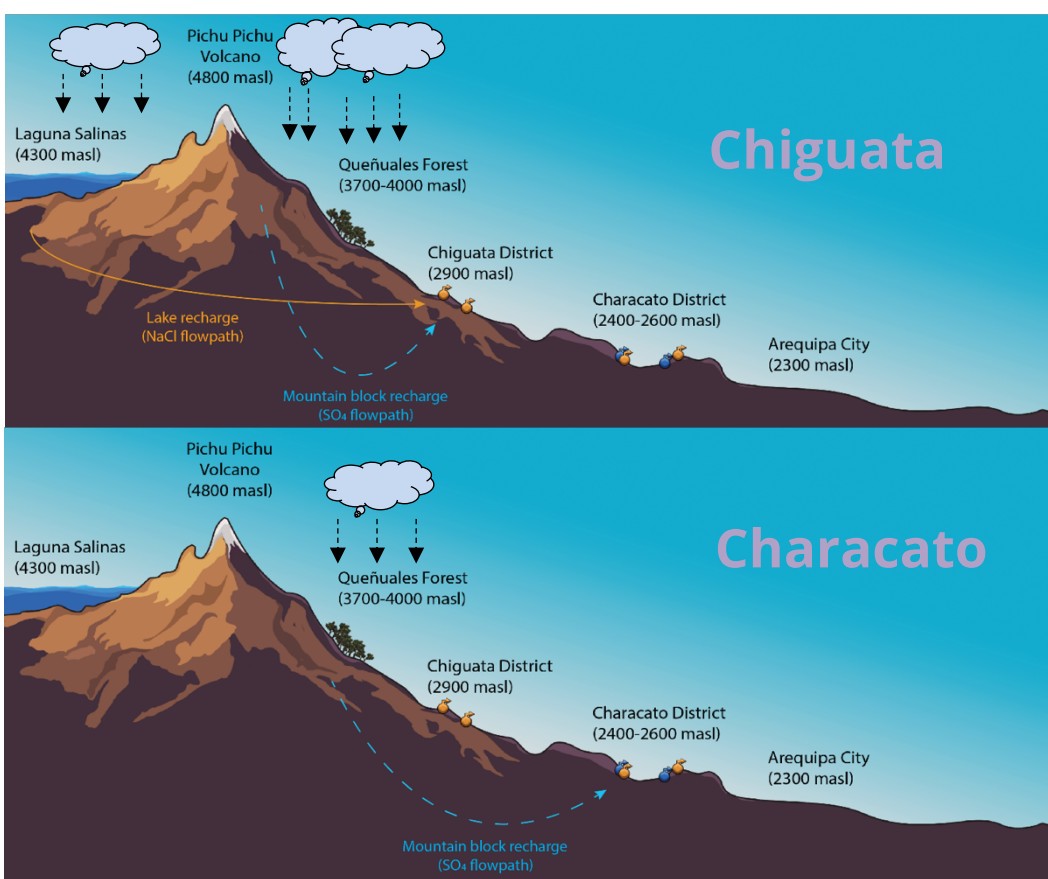

**Figure 11.  Cross section of our study area in the Quilca-Vitor-Chili basin, east of the city of Arequipa (credit: Diego Alvarez). top)**
**Conceptual model identifying zones of recharge and flowpaths in Chiguata and bottom) Characato. Note, the conceptual groundwater flowpaths do not cross in the top panel.**

Our explanation for the isotopically light and Cl- and B-rich source of recharge is that groundwater flowpaths originating in the surrounding mountains converge beneath the salar and subsequently flow through fractured volcanic rocks until these

flowpaths are intercepted by the fault zones along the Río Andamayo. Closed-basin salars like Laguna Salinas often contain thick, clay-rich lacustrine deposits at the bottom of the salar which can impede vertical flow. Very little, if any, recharge must





occur in the Laguna Salinas basin during the dry season when evaporation rates are highest or else the isotopic content of the recharge would show the effects of evaporation. Deep groundwater flowpaths, in comparison, would preserve the light isotopic signal of high-elevation recharge. It seems plausible that the Laguna Salinas basin recharge is a mixture of groundwater

flowpaths that originate in the surrounding mountains, converge beneath the salar (local topographic low), and mix with wet-season recharge in the salar (leakage from the lake). Ultimately, the isotopic variability of precipitation and mineralogy of Tacune and the surrounding mountains located north of the salar are poorly quantified. Future studies should: 1) investigate the vertical hydraulic conductivity of these lacustrine sediments and quantify their spatial variability in Laguna Salinas, 2) quantify the spatial variability and elevation-dependency of precipitation and the stable isotopic composition of precipitation

in the mountains surrounding Laguna Salinas, 3) quantify the lithology and mineralogy of the surrounding mountains, and 4) install equipment to close a water balance on the closed basin.

## 6 Conclusions

One of the goals of our research was to elucidate the source and recharge zones and groundwater flow paths contributing to vital lower elevation springs located east from the city of Arequipa, Peru. Our work was able to identify two main sources:

mid-to-high elevation snow/rain from the Pichu Pichu volcano and infiltration through the salar sediments of Laguna Salinas which mix in different proportions in geographically-distinct spring clusters. While the queñuales forest in the Pichu Pichu Volcano is the main zone of recharge for springs in Characato, inter-basin groundwater flow from Laguna Salinas recharge has a major influence on springs in Chiguata. This means that changes in rain patterns in the high-elevation closed-basin Laguna Salinas as well as changes in rain and snowmelt in Pichu Pichu Volcano have an impact on the maintenance of these

lower elevation springs. The potential importance of the queñuales forest for groundwater recharge of Characato springs also highlights the need to consider careful management of this natural ecosystem. The presence of inter-basin groundwater flow from the Laguna Salinas closed basin may further complicate the sustainability and management of the springs by making them dependent on a regional groundwater system, as opposed to local groundwater. Thus, water resource planning including the Chiguata springs, one of which is a main drinking water source to the city of Arequipa (La Bedoya spring), must consider

the influence of changes in land use not only at the queñuales forest, but also at Laguna Salinas where mineral extraction companies currently operate. While our findings are specific for this region in southern Peru, the implications of climate change, high elevation forest ecosystem degradation, and influence of the relatively unexplored factor of IGF to low elevation springs in the Central Andes may translate to other regions of the Andean plateau and Western Cordillera.

While Laguna Salinas is a topographic closed basin, some proportion of surface water from the salar infiltrates the lacustrine sediments, recharges beneath the salar, mixes with fresh mountain-block groundwater, and is incorporated into the regional groundwater flow system. This finding confirms that high-elevation salars may be important sources of recharge to low-



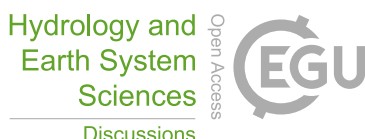

elevation springs, and calls attention to careful consideration of connectivity between other high-elevation salars and low-elevation springs in future hydrogeologic studies.


Our research informs the origin and flowpaths of groundwater resources in this region; however, several questions remain. For instance, what is the percent contribution of each identified recharge zone to spring maintenance and what is the contribution of rain versus springs to river flow? Water managers would greatly benefit from learning the answer to these questions in order to focus their management and conservation efforts of both spring and river waters that are of high importance to socio-
economic activities of the region. The combination of isotopic and chemical-based tracers with traditional discharge monitoring, subsurface hydrogeologic mapping, and monitoring of hydraulic head data may be useful techniques to address these questions. However, the use of the latter becomes difficult in remote, poorly accessible areas such as those found in many areas of our study. Finally, another question that remains is the presence and extent of pre-modern groundwater recharge in this previously glaciated region and the potential influence of paleorecharge to low elevation spring maintenance, which would
have significant implications for sustainable water management in this region. In this arid region of southern Peru, water resources research should continue to investigate the processes that control groundwater and surface water sources, and how to best protect and manage them for future use.

**Acknowledgments**

This work was funded by the Universidad Nacional de San Agustin (UNSA), Peru through the Arequipa Nexus Institute for
Food, Energy, Water, and the Environment. The project director Dr. Victor Maqque was instrumental in nurturing this international collaboration. We thank Alexandra L. Meyer and Janine M. Sparks for their assistance in the stable isotope laboratory.

**Author contribution**

OAC: Data collection, laboratory analysis, data analysis and interpretation, and drafting manuscript. EJO: Data collection,
laboratory analysis, experiment design, data interpretation, and manuscript revision. MDF and LRW: Experiment design, data interpretation, manuscript revision, and secured funding for the project. SAZ: Regional geologic knowledge, data collection, manuscript revision. JDR: Regional geologic knowledge, manuscript revision. WRQ, CSM, and MAC: Data collection, data reporting, and manuscript revision. Final version of the manuscript was approved by all authors.

**Competing interests:** The authors declare that they have no conflict of interest.



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

isotopes in the Andean Coastal Cordillera reflect the transition between Atlantic and Pacific moisture influence sub-
        seasonally, *in review at GRL*.