# Peer review of "Evidence for high-elevation salar recharge and interbasin groundwater flow in the Western Cordillera of the Peruvian Andes"

_Hydrology and Earth System Sciences, 2021_

## Author Comment (AC1)

**Reviewer 1 comments (response in red)**

**General Comments**

This is an interesting study that helps inform our understanding of recharge, groundwater flow, and spring origins. However, it needs attention before it is suitable for publication.

Firstly, the paper needs better focus. Some sections (e.g. the discussion of geochemical processes) do not seem to inform the overall story and are a distraction. The description of the study area is long and some of the information does not seem to be relevant. There are also descriptive sections in the discussion and a tendency to repeat information. A more succinct tightly-written paper would be easier to follow and have more impact.

More importantly, in several places the discussion of processes is not convincing (inc. the geochemical processes, groundwater residence times, and lapse rates). The explanations tend to be long and are sometimes not consistent or are overly speculative. The central point that the springs are recharged at altitude and the water follow different flowpaths appears reasonable, but this sometimes is lost.

Finally, the Conclusions are mainly parochial . Some idea of the broader significance of this study or comparisons with similar environments would give the paper a broader appeal.

I hope that these comments are useful.

We thank for the reviewer for the thoughtful feedback and suggestions for making the manuscript more concise. Most of the comments can be addressed during manuscript revisions. Detailed responses follow below. We apologize for the inconsistent verb tense throughout. Some changes have already been incorporated into the manuscript, though the revision has not been invited by HESS.

**Specific Comments**

*Abstract*

The abstract gives a good idea of the major conclusions of the paper. However, it needs more focus on the results and conclusions rather than the aims. You should report a

few key values in the Abstract as qualitative descriptors ("stable", "higher" etc) do not convey much specific information.

The review makes good suggestions we can address in a revised manuscript.

*Introduction*

The introduction is clear and well structured. The first paragraph (lines 40-55) would benefit from a few extra details if these are available, specifically:

- Are the impacts on groundwater resources quantified? Groundwater resources in this region are relatively poorly quantified. There are substantial knowledge gaps with respect to hydrogeology in the region and our paper attempts to address a subset of these knowledge gaps.

- Likewise are there estimates for how much the recharge may decline? No, this is difficult without 1) some baseline assessment on how much recharge occurs historically in this region, 2) some assessment on the sources of recharge (provided in our paper), and 3) some information on how regional climate change will impact the source of recharge (e.g., changes in rainfall, changes in high-elevation snowfall, loss of alpine glacial cover). These data are lacking.

This would help with understanding the context of this study.

Lines 57-65. Provide some references for this material. References added.

Lines 73-79. This is only true is there is no other mechanism for exporting Cl. In some saline lakes Cl is lost via salt deposits being eroded by the wind (deflation). Is that the case here? If the lakes do recharge the local groundwater system, I would expect there to be some evidence of that (shallow high salinity groundwater around the lakes) – any evidence of that? The reviewer's point is noted and we can add wind erosion as a potential mechanism to remove salts from the salar. The studies that we cite in this introductory section have not come to that conclusion though. With respect to high-salinity shallow groundwater in the salar, wells are sparse in this area and we do not have access to wells used by mining companies. To the best of our knowledge there are no biological surveys indicating the presence of basin fringe halophytic plants that can be used to mark the spatial extent of shallow saline groundwater. We will be sure to discuss the salinity in the wells we did sample near the salar in this study.

*Study Area*

This is comprehensive, but in places the descriptions are lengthy.

- There is some repetition with the introduction (eg lines 119-129). The information provided in this section is specific to our study area and highlights the importance of La Bedoya spring and the intensity of agricultural and mining activities in this region. We condensed the material to these main points.

- Some of the details of the geological history (lines 137-148) seem superfluous. There are only 2 sentences that reference the geologic history, but we can trim slightly.

- The climate description (Section 2.3) could also be more succinct – the important details are a bit lost in the narrative. We can cut the description of recent trends which are not particularly relevant to this study and condense that idea in the introduction motivation.

- Again with Section 2.4, how much of this detail is really necessary

  We can focus on describing the specific low and high permeability geologic units and the faults located in the region. We can describe the salar but save the geochemical description of the lacustrine deposits for the discussion when they are most relevant to the reader.

The lengthy descriptions mean that it is not always clear what the important points are. Try to give more focus to what is important for the study rather than trying to cover everything. We believe specific comments related to this theme are addressed below.

The order is also not intuitive – Sections 2.2 and 2.4 both deal in part with the Geology and Geomorphology. It would be better to merge these two (or at least make them sequential) and to present the material in a large- to smaller scale order. So the descriptions of the large-scale geology (lines 200-207) would be better presented before the description of the basin (Section 2.2).

Sections 2.2 and 2.4 can be reduced in length and reorganized as suggested in a revised manuscript.

Figures 2 & 3 can be merged – they are maps of the same area. That way it would be easier to see the relationships between the samples and the faults. Also, it is not currently clear from Fig. 2 what is sampled (springs, surface water etc) without referring to the Table – make the symbol for each different.

We can make these suggested figure changes in a revised manuscript.

*Methods*

Here also there are some diversions (eg Lines 261-265) that are distracting and do not belong in the methods. If this information is important, then it belongs in the study area section.

The additional descriptions of sampling locations were moved to the study area section.

Lines 270-272. Really it is only SpC that is measured (This is not correct. It is only the electrical conductivity that is measured. Specific conductivity and TDS are calculated from the electrical conductivity), the TDS is just calculated from the SpC using an assumed conversion. If you have a full suite of ions (which it looks like you do), you can calculate TDS as the sum of the ions. If you stick with the TDS estimated from SpC, you need to note that and provide the conversion. We can provide the conversion factor.

Section 3.2. A few more details on the Tritium procedure (enrichment and equipment) needed

The tritium analyses were not performed in house, but we will add additional details from the analytical lab.

Section 3.3. Need to report precision for the major ions

This was added.

*Results*

As explained below, this section loses focus in places and it is not always clear what the important points are. You need to be clearer as to what information is important for this study and concentrate on that. The figures need improvements -duplicating the figures just to show the two Salinas samples is excessive. It is clear that these are evaporated surface waters and you do not say much more about them than that. In that case just omit them from the figures and note that in the caption.,

H isotope ratios should have no decimal places to be consistent with the quoted precision (O is fine with one decimal place)

We can make these suggested changes in a revised manuscript.

Lines 314-317. Report the low 3H activities as bd without the +/- (which have no meaning for 3H below the lower detection limit). If the precision is 0.1 TU (Section 3.2) then the +/- of 0.04 for BED is overoptimistic. Given that this is very close to the lower detection limit, how sure are you that this is real. Given that you sampled the springs at their surface outlets, could it just be a small amount of modern water (eg recent rainfall) mixed in with the spring water?

We state in the manuscript that water samples at or below the detection limit are tritium dead. However, we can make a note stating that the samples had 3H values that were below the detection limit and we can remove the lab-reported numeric values. In any case, we think that it is sufficient to call these samples tritium dead.

Figure 4. The two diagrams are a little confusing as they present the same data. If you feel that you need 4b to show the detail, you could make 4a in inset with just the Salinas surface water on it. Also need to reference the LMWL in the caption.

We can make Fig 4a an inset in Fig 4b and reference the LMWL in the caption.

Figure 5. Again, you do not need both figures – you can just use Fig. 5b and explain that you have omitted the Salinas waters. It would be useful to have Fig. 4 & 5 as a single figure as it would show the relevant stable isotope data in one place.

We can take the reviewer's suggestion, remove Fig 5a and combine current panels 4b and 5b into one figure.

Section 4.3

This section lacks focus. What is important here (the processes or the differences in chemistry). The discussion of the processes using the Gibbs Diagram is not very convincing and could be done better. however, consider whether that is important. If the important point is that the waters have different geochemistry and so follow different flow paths, then just show that (the Piper and/or a couple of bivariate plots like Fig. 10, and a brief description would suffice). The discussion of processes is not very clear and may not be necessary.

The Gibbs Diagram in particular is not very informative in determining processes and you have the data to do that more rigorously. Waters dominated by evapotranspiration have Cl/Br and Na/Cl ratios close to those of rainfall (which you can probably estimate). Extensive rock weathering produces high cation/Cl ratios while halite dissolution produces very high Cl/Br ratios. A few bivariate plots (eg Na/Cl vs. TDS and Cl/Br vs. TDS) would show that much better than the approach that you are currently using. There is extensive literature on this (eg numerous groundwater papers by Mike Edmunds).

The Gibbs diagrams were used to show that flowpath groups fall within the expected processes within the diagram (eg. Salar waters within evaporation and bedrock springs within rock weathering). We also reference the paper by Marandi and Shand (2018) which uses an example from Edmunds et al. (1987) to show that groundwaters which plot in the evaporative enrichment region of the Gibbs diagram may not have experienced dissolution of evaporites. In the case of Edmunds et al. (1987), the groundwaters evolved toward this Na-Cl type water due, in part, to mixing with saline connate groundwater. Thus, an argument can be made against the Gibbs diagram that false evaporation processes can arise in geologic settings where marine deposits and saline groundwater are present. In our study area there are no ancient marine deposits that would conflate our interpretation of the salar waters as originating from evaporative processes. The Gibbs diagram clearly shows that the low-elevation spring samples which appear to be receiving some mixture of high-elevation high-salinity recharge fall along a proposed mixing line with this subspace. This does not appear as clearly in the plots suggested by reviewers (provided below). The Cl/Br plots are also problematic because few samples had detectable levels of Br and so this data set is much smaller than that shown in the Na/Cl and Gibbs diagram.

In addition, Marandi and Shand (2018) state in their critque of Gibbs DIagrams that, "It remains possible that recharging waters cross the water table with a chemical composition plotting in the upper right corner of the Gibbs Diagram, perhaps particularly in regions where land uses have caused salinization of soils or other artificial impacts to near surface mineralogy." This is the exact condition which we propose is happening in the connection between saline recharge from the salar and the Chiguata springs. Saline recharge would plot in the evaporative enrichment field where the Laguna Salinas surface waters plot. This saline recharge then mixes with non-saline groundwater flowpaths in the mountain block that were recharged by high-elevation snow and/or rain. Thus, Chiguata springs plot between the fields for evaporative enrichment and precipitation dominance. In comparison, springs such as the Quenales Forest springs and Pichu Pichu springs, which are supported by high-elevation recharge from snowmelt and/or rain, plot near the precipitation dominance

field for anions and rock-weathering for cations. Their placement in these fields make physical sense.

We will restructure and streamline this section to make these points clear. We are not convinced that adding other bivariate plots will necessarily change this story. For example, we plotted Cl/Br mass ratios and these essentially show the same result. Saline recharge has high Cl/Br and relatively low TDS because high-elevation springs we sampled flow into the salar and the salar refills during the rainy season with dilute runoff from surrounding mountains. As this saline recharge mixes with other non-saline groundwater flowpaths, the Cl/Br ratios tend to decrease while the TDS increases due to rock-weathering reactions. This interpretation is not different from that inferred from the Gibbs Diagram.

[Figure]

[Figure]

Lines 386-394. I presume that this also shows up in the other parameters? I'm not sure that you have enough data to do a PCA or cluster analysis but you should make the point with the other parameters.

We do not think it's appropriate to use a cluster analysis on this dataset. Figures 8 and 10 supports this interpretation as well. We can edit the discussion to summarize all evidence of mixing multiple groundwater flow paths in the discussion.

*Discussion*

The Discussion covers a range of topics but there are a number of potential inconsistencies and unclear explanations. This is the most important part of the paper, so more clarity would help. I also suggest that you add a couple of sentences to the start of the Discussion as a guide to what you will be dealing with.

This is a good suggestion that we plan to incorporate.

Sections 5.1.1 & 2.

The discussion regrading the lapse rate (lines 415-434) needs to be clearer. While it is true that springs can have stable isotope ratios that vary with altitude that is mainly the case where they are recharged close to where they discharge. In the case of your springs, you make the case (Section 5.1.2) that they are recharged at high altitudes. That interpretation is reasonable. However, the way that this discussion is presented is to set up the idea that the springs should vary with altitude (line 416) and then point

out that that is not the case and then interpret the data in terms of recharge altitude in Section 5.1.2.

We can edit the framing to make it easier on the reader.

Much of Section 5.1.1 (the correlation with altitude etc) contains observations that should be part of Section 4 – some of it is in there already and it is just repeated here.

We do not think any new results are presented in section 5.1.1 other than 1 sentence reminding the reader that the isotopic composition of rain at the lower elevation matches the spring water at the higher elevation. We can cut that. The rest of this section provides discussion and context for the observations reported in Section 4.

The magnitude of the lapse rate.  If the springs are recharged at higher altitudes then you can't use them to estimate the lapse rate. In that case, your lapse rate should be based on the surface water samples.

We agree with the reviewer's point that springs recharged at higher elevations will be more isotopically depleted than precipitation at the same elevation as the spring emergence unless the flowpaths are very short or storage is small. The curious thing in this case is that the springs at a HIGHER elevation are isotopicaly similar to precipitation at a LOWER elevation. Since we do not claim that lower elevation rain is recharging higher elevation springs (against gravity), we can conclude that precipitation at the higher elevation spring recharge area is similar to the lower elevation precipitation we observed.

This led us to consider the inferred precipitation isotope lapse rate in this section of the mountain, ~0 permil per 1.4 km. If the precipitation at the elevation of the Quenuales forest were much more depleted than Characato elevation, there is no way to explain the isotopic composition of the mid-elevation Quenualues forest springs. This is probably an unusual situation that confused the reviewer. Also note that we are discussing lapse rates across 1.4 km of elevation change in one section of the mountain, not following dominant wind patterns. When considering the entire mountain, the highest elevation snow samples are much more depleted than the lower elevation. We will include the lapse rate calculated over the entire elevation we have data for in the revision and highlight the anomaly between the Characato and Quenuales elevations within that larger gradient. We also have additional data from the 2020 rainy season that we present in one of the following comments. More importantly, we will edit this section to make the main focus on the recharge elevations inferred from the isotopic data and not an analysis of the isotopic lapse rate itself.

However, those data may not be suitable, specifically:

- Rainfall sampling is referenced to an unpublished study. It is not clear how many samples this represents and the duration of the rainfall record. Given the likely variability of rainfall isotope values, you ideally would have a multi-year weighted average value, but is this the case?

This is true. Precipitation collection was initiated for this project and included daily rain collections during the 2019 rainy season (effectively 1 year because there is only trace rain in other months). In a revision, we can now also include additional data generated after this manuscript submission that shows 2020 precipitation collected at both Characato and Chiguata were isotopically similar to each other, but also that they were less negative than the 2019 precipitation collected at Characato.

Characato 2019 (-9.12,-63.51); Chiguata 2020 (-6.3,-38.1) ; Characato 2020 (-5.5, -31)

- Snow is probably mainly winter precipitation and is difficult to use with samples that represent long-term averages.

We are unclear what the reviewer's point is here. The snow on Pichu Pichu occurs during the summer. Summer is the only season with measurable precipitation with rain at lower elevations and snow at higher elevations. Seasonal bias is not a complication in this system.

- The surface water samples seem to be partially fed by spring water (Section 5.2). If those springs were recharged at high altitudes then using river water to calculate lapse rate is possibly not valid as it is not capturing only rainfall at the altitude where you sample it.

We did not use the surface waters to calculate the lapse rate.

- A similar concern would apply to any rivers that flow from high to low altitudes and thus mix rainfall from a variety of altitudes

Agreed. See earlier comments.

- Even if the rivers are mainly fed by local rainfall, their stable isotope values are likely to vary seasonally (as you discuss in Section 5.2) and so again are difficult to use in this way.

Agreed. See earlier comments.

I'm not convinced that you can determine the lapse rate with the data that you have. If you are going to include this discussion, it needs to be more convincing. Otherwise you may be able to estimate it using other studies?

The main goal is to estimate the recharge elevation of the lower-elevation Characato springs. These lower elevation springs are isotopically similar to the mid-elevation Quenalaes springs. We interpret the recharge zone of the low elevation springs as the same elevation of the Quenalaes spring recharge. This is the most important observation. The discussion of the inferred precipitation isotopic lapse rate may have created a distraction for the reader.

Regardless, the biases that the reviewer mentions above are avoided in this specific case. If we consider Pichu Pichu snow, there is a substantial isotopic gradient over the region. The comments in section 5.2 are specific to the 1.4 km between Characato and the Quenuales forest on one directional slope. We agree with the reviewer that interpreting lapse rates from surface waters to determine recharge elevation would be problematic. That is not what we did. There are very few precipitation isotopic measurements in this region. In a revision, we could also include additional data generated after this manuscript submission that shows 2020 precipitation collected at both Characato and Chiguata were isotopically similar, but also that they were less negative than the 2019 precipitation. This could be used to show that the Characato precipitation is variable year to year, but that it could be more enriched compared to the Characato spring observations which do not vary and we infer are recharged from higher mid-elevations.

Characato 2019 (-9.12,-63.51); Chiguata 2020 (-6.3,-38.1) ; Characato 2020 (-5.5, -31)

Spring recharge elevation. This seems broadly correct; however, it becomes more doubtful if there are palaeowaters in the basin (lines 463-465). Discharge of paleowaters into surface water bodies also complicates the lapse rate calculations. Is there anyway to test this idea? While you do not have radiocarbon data are there examples of palaeowaters in analogous settings or examples of nearby springs for which there are better residence time calculations?

We do not have evidence of paleowater discharge from this basin, but since the region has a glacial history, we cannot absolutely rule it out. We thought it best to mention this possibility, but can remove this mention if the reviewer thinks it's unnecessary.

Section 5.3 also needs attention.

- It is not clear where the 3H activity of "young aquifers" of 2.9 TU comes from

This came from recharge-weighted well-mixed aquifer mixing models informed using time series from Albero & Panarello (1981) - this includes a compilation of time-series from South America. High-elevation snow was measured in this study and it had a tritium activity of 2.2 TU. A tritium activity of 2.9 TU is not beyond the realm of possibilities based on our measured data. In fact, we commonly find that

tritium activities decrease with decreasing elevation (with inferred increasing flowpath length, degree of mixing, and storage). Manciati et al (2021) show a mean precipitation activity of 3.0 TU near Quito, Ecuador which is admittedly north of our study site, but not disparate from our measurement on Misti Volcano. In any case, we can restate that we think high-elevation mountain-block aquifers are "relatively young" based on our tritium activity for Misti and cite recent research in the western U.S. T

- The definition of fossil water as being >60 years is largely a northern hemisphere viewpoint as the higher 3H bomb pulse waters are still detectable in groundwater. This is not the case in the southern hemisphere where the bomb-pulse tritium has decayed back to natural levels (e.g., Morgenstern, U., Stewart, M.K., Stenger, R., 2010. Dating of streamwater using tritium in a post nuclear bomb pulse world: Continuous variation of mean transit time with streamflow. Hydrology and Earth System Sciences, 14, 2289-2301; Tadros, C.V., Hughes, C.E., Crawford, J., Hollins, S.E., Chisari, R., 2014. Tritium in Australian precipitation: A 50 year record. Journal of Hydrology, 513, 262-273).

  We provide data from Moran et al. (2019) where they define "fossil water" as being greater than 60 years. Please note our parentheses in this statement. We do not call groundwater in this manuscript "fossil groundwater" because we do not necessarily think that fossil groundwater is as young as 60 years. Tritium-dead groundwaters are not unique to the southern hemisphere; this is increasingly a problem in the northern hemisphere. "Tritium dead" is admittedly an ambiguous term, as we state in the manuscript.
  Rather than use tritium data from Australia, which is well documented, we wanted to place our tritium data in the context of the tritium breakthrough in South America. We specifically state in the manuscript that, "The atmospheric breakthrough of 3H associated with nuclear weapons testing in the 1950s and 1960s for South America peaked at approximately 60 TU in 1965 (Albero & Panarello, 1981)."

- The residence time of 300 years seems arbitrary. Presumably it is based on mixing at the top of the aquifer but you have a fractured flow system that is likely to behave very differently.

This is based on recharge-weighted well-mixed aquifer models that were informed using time series from Albero & Panarello (1981) and some back-of-the-envelope calculations of specific discharge. The mixing models have high uncertainty because these models are sensitive to the fraction of recharge that is used to weight the tritium annually. We think that these springs likely have

residence times falling between the ranges for tritium and radiocarbon. The flow calculations are uncertain because the region is data poor and we do not have well-constrained hydraulic parameters. Therefore, we will simply state that the groundwaters are older than 60 years until additional data become available.

As you have only three tritium measurements (all of which are close to or below detection) and you do not have a good idea of the rainfall values (2.5 to 10 TU is a large range), there is little quantitative that you can say here and this section is not that informative. I'd just make a case for the water being at least a few decades old in Section 5.4.

We also provide tritium data on modern snowfall collected at high elevations on Misti Volcano and in rainfall at low elevations. Although our samples are limited in number (funding was not sufficient to do a broader scale sampling campaign), modern precipitation has tritium activities ranging from 1.4 to 2.2 TU. This range is much smaller than the other data which we provide for context only. We will state explicitly that this data was provided for context only to avoid confusion. In any case, we would expect "young aquifers" at elevations greater than the elevation of the salar to have measurable tritium. The springs emerging above the salar are tritium dead. We can conservatively interpret those observations as at least several decades old.

Section 5.4

Lines 517-520. I'm not sure that I'd expect Cl to increase along flow paths. To do so implies that Cl needs to be added from the rock matrix as evapotranspiration is a surface process. That will only occur if there is halite in the rocks (which is not that common). This concept does appear in may textbooks but the supposed process is never really explained.

Cl concentrations should only be invariable along flow paths if: 1) piston-flow processes are invoked and groundwater doesn't mix with other flowpaths, or 2) if Cl is not removed or added by dilution or biological processes. We propose that mixing occurs at this large spatial scale within a fractured mountain block. We can cite Bresciani et al. 2018 for this.

Lines 517-525. You look to have measured Br. Cl/Br ratios will readily determine whether you have halite dissolution (Cartwright, I., Weaver, T.R., Fifield, L.K., 2006. Cl/Br ratios and environmental isotopes as indicators of recharge variability and groundwater flow: An example from the southeast Murray Basin, Australia. Chemical Geology 231, 38-56). You may not need to speculate here.

Br was not detected in all samples, but below is a plot of samples with Br measurements from Chiguata and the springs feeding into Lagunas Salinas. The plot below is consistent with halite dissolution influence in the Chiguata springs. The Lagunas Salinas springs are sampled before they interact with the salar water.

[Figure]

[Figure]

Section 5.5.

The first paragraph (lines 569-574) repeats the previous section and is not needed. OK

Figure 11 only needs one panel as you can show all three flow paths without confusion.

We can merge the information in this figure in the revision.

*Conclusions*

Again, there is some repetition here. Instead of repeating the specific findings, which you cover in Section 5, try to outline the general points. However, you should explain the general importance of the study or compare it with similar studies elsewhere. This will make the paper appeal to a wider readership.

We can shorten the summary information and focus on the broader impacts.

Two general points we can emphasize are:

1) Results from our study show evidence of this high elevation closed basin salar is hydrologically connected to lower elevation springs. We place this in the context of other salar studies in the Andes and conclude that not all salars are evaporation pans, but as other studies have found, some leakage through salars recharges regional groundwater.

2) This study provides perspective on the long-term stability of groundwater in this arid region. The tritium data as well as repeat discharge and stable isotope measurements supports long-flow paths with residence times of water being at least a few decades old if not longer. In the face of climatic warming these data suggest that regional groundwater may provide a stable water resource over the next few decades.

References:

Albero, M.C. and Panarello, H.O.: Tritium and stable isotopes in precipitation water in South America, 1981

Bresciani, E., Cranswick, R. H., Banks, E. W., Batlle-Aguilar, J., Cook, P. G., and Batelaan, O.: Using hydraulic head, chloride and electrical conductivity data to distinguish between mountain-front and mountain-block recharge to basin aquifers, Hydrol. Earth Syst. Sci., 22, 1629–1648, https://doi.org/10.5194/hess-22-1629-2018, 2018.

Edmunds, W. M., Cook, J. M., Darling, W. G., Kinniburgh, D. G., Miles, D. L., Bath, A. H., Morgan-Jones, M., and Andrews, J. N.: Baseline geochemical conditions in the Chalk aquifer, Berkshire, U.K.: a basis for groundwater quality management, Applied Geochemistry, 2, 251–274, https://doi.org/10.1016/0883-2927(87)90042-4, 1987.

Manciati, C., Taupin, J. D., Patris, N., Leduc, C., and Casiot, C.: Diverging Water Ages Inferred From Hydrodynamics, Hydrochemical and Isotopic Tracers in a Tropical Andean Volcano-Sedimentary Confined Aquifer System, Front. Water, 3, 597641, https://doi.org/10.3389/frwa.2021.597641, 2021.

Marandi, A. and Shand, P.: Groundwater chemistry and the Gibbs Diagram, Applied Geochemistry, 97, 209–212, https://doi.org/10.1016/j.apgeochem.2018.07.009, 2018.

Moran, B. J., Boutt, D. F., and Munk, L. A.: Stable and Radioisotope Systematics Reveal Fossil Water as Fundamental Characteristic of Arid Orogenic-Scale Groundwater Systems, Water Resour. Res., 55, 11295–11315, https://doi.org/10.1029/2019WR026386, 2019.

---

## Author Comment (AC2)

**Reviewer 2 comments (response in red)**

The paper "Evidence for high-elevation salar recharge and interbasin groundwater flow in the Western Cordillera of the Peruvian Andes" by Alvarez-Campos presents a multi tracer (isotopic and geochemical) assessment on the influence of groundwater flowpaths from a close basin salar in spring water upwelling at lower elevations that supply water to the city of Arequipa, Peru. Overall, I find the paper well-structured and clearly written and the findings generally well supported by the presented data and analysis. My major concern relates to the insufficient description of sampling collection and laboratory analyses. Given the relevance of the paper for the management of water resources in the region of study, I consider it is suitable for publication in HESS after some points described below are implemented in the manuscript.

We thank the reviewer for highlighting methodological details that were missing and can add them to the manuscript.

Major comments:
L111-113: tritium and residence time come as a surprise for the reader. I suggest adding a few statements or a short paragraph to the introduction mentioning the value of tritium in the context of the study, highlighting particularly research on the study region for similar purposes.
We can add a short paragraph introducing the value of tritium in the context of this study in the revised manuscript.

Section 3.1. It would be helpful to include the elevation of the sites for reference in this section so the reader does not need to check Table 1 many times. Also, it is confusing that the authors sometimes mentioned only the names of the sites, other times only the sites IDs (presented in Table 1), and others, both names and IDS. I strongly suggest to homogenize this in the whole manuscript, figures and tables for consistency and clarity (i.e., this issue is common in this and the rest of the paper sections).
This is a helpful suggestion to improve readability of the manuscript.

L260: describe how snow was sampled.
Detail on snow sampling were added.

L261: report the period and frequency of rainwater sampling. If not collected throughout the whole study period, indicate why. I strongly suggest to give a name to the precipitation water sampling in Table 1 and use it in the whole manuscript. Also, show it and add it to the legend in all relevant maps.
Rain was sampled over the monsoon season January-March. The rest of the year there is no precipitation to sample. We can include this clarification and give it a site ID.

L267-272: the description of water sampling collection is quite incomplete and requires substantial improvement. Some of the main issues are: how was water from river and springs samples? How were samples collected for stable isotope analysis stored to avoid fractionation by evaporation? How was rainwater sampled to assure evaporative fractionation did not affect the water samples? What sits were samples for tritium, specify? Report the made, model, and accuracy of devices used to measure physico-chemical parameters in situ and how often and how they were calibrated. Please update the paragraph with this and other relevant information that might be missing.

We detail in the methods isotope section 3.1 how samples were collected, storages and sealed to avoid evaporation. We have added further details on rainwater collection and evaporation. Samples for tritium are listed now in the methods section. In situ calibration and instrumentation data are included now for the Oakton pH meter and daily sensor calibration in the field.

L286: I am puzzled about the construction of the LMWL using data for 3 months only. In section 2.3, it is mentioned that the very dry winter occurs between June and August, however, it is not clear if precipitation during those months is at all nonexistent, or just very little compared to the wet summer monsoon one (November to April). Even for the latter, using an isotopic dataset from January through March 2019 might not be entirely representative of the local conditions. I strongly suggest the authors to include a time series of precipitation during the study period in the paper for reference, and discuss if and how the limitation of the available isotopic dataset could influence their findings. Showing the precipitation amount data could also help to link their findings about the influence of modern day recharge on their findings and the developed conceptual model.

It only rains a few months out of the year, during the summer (~Dec-Mar). Our 2019 precipitation sampling is effectively an annual summary. For the reviewer's reference, the LMWL calculated for the 2019 wet season was y = 8.01x + 8.40. Including 2020 precipitation, the fit was very similar, y = 8.05x + 10.91. We would be happy to include the monthly rain amount data to illustrate the seasonal rainfall in this location. However, we do not base any of the conclusions or analysis in the paper on the LMWL. It is only provided for general context. A separate manuscript is submitted that analyzes the daily precipitation water isotope data from an atmospheric process perspective. Adding it here would add unnecessary detail to an already lengthy paper.

Section 3.3: there is very little information about the chemical analysis. Please report standards, calibration curves, detection limits, etc. used for the analysis of anions and cations. Also report QA/QC procedures to secure high quality of the produced data.
These can be added.

L.346-350: I strongly suggest to show the data supporting these statements (i.e., similarity between spring and surface waters). One option is to have a subplot in Figure 6 showing the springs' isotopic compositions. It would also be good to include the isotopic

composition of precipitation in such a plot (e.g., adding a third panel, or plotting together with the springs and surface water isotopic fingerprints?)

L.447 and L.502: how do the authors infer that residence time should be several hundred years old? If anything, based on the Tritium dead results, one could say that groundwater is older than ca. 60 years based on the 1960s bombings. However, without further evidence, saying that water is of certain age seems arbitrary and could be misleading. The authors might be right, but further discussion is needed to justify their statement. Otherwise, please recognize the limitations of the presented dataset and do not speculate about water aging. Based on this comment, I strongly suggest the title of section 5.3 is updated to "Insights into groundwater age" or something similar since results presented are not conclusive.

This is based on recharge-weighted well-mixed aquifer models that were informed using time series from Albero & Panarello (1981) and some back-of-the-envelope calculations of specific discharge. The mixing models have high uncertainty because these models are sensitive to the fraction of recharge that is used to weight the tritium annually. We think that these springs likely have residence times falling between the ranges for tritium and radiocarbon. The flow calculations are uncertain because the region is data poor and we do not have well-constrained hydraulic parameters. Therefore, we will simply state that the groundwaters are older than 60 years until additional data become available.

Light/heavy versus enriched/depleted: throughout the manuscript, the authors use these terms interchangeably. I strongly suggest the authors to avoid using the terms light/er when referring to depleted isotopic compositions to avoid confusion with the commonly used isotopic terminology of light (more abundant) versus heavy (less abundant) isotope ratios. Please revise the whole manuscript to make changes accordingly.
We can make these changes and remove informal usage of light/heavy.

Minor comments:
L50-55: Please support these statements with appropriate references.
L65: add references to support the final statement of the paragraph.
We can add additional references.
L67: this is not true for the whole western South America because i) the northern (tropical) Andes in the north are generally humid and salars are mostly common in areas of the central Andes. Please specify the particular region of the Andes for which this statement applies in the whole manuscript.
We can make this clarification.
L96: report elevations of the Laguna and salar
This information was added
L145-147: add references for the statements in these lines
References can be added.

L159: similar to L67, specify the specific region across the Andes for which this statement applies.

We can make this clarification.

L170: specify which rivers

This information was added

L173: from 2018 to ??? please specify

This sentence was modified to specify that this is annual 2018 precipitation data.

L196: report values of the predicted precipitation decrease

L227-232: Misti and Pichu Picu volcanic complexes are quite relevant for context. It would be super helpful to show them in Figure 3.

We can add labels.

L261: report names (or IDs) of the sampled springs

Sample IDs were added in the text

L.282: add references for memory effect on isotopic analysis

OK

L.290-292: six sampling sites are listed here, whereas only four sites were mentioned in section 3.1. Please clarify. Also, please report the instrument and standards used for tritium analysis

The four sites mentioned correspond to samples analyzed for tritium, which were snow from Pichu Pichu Volcano, springs from Laguna Salinas, one spring from Characato, and one from Chiguata. Section 3.1 indicates that we sampled six springs. Four of these springs obtained in the district of Characato, and two other springs sampled in the district of Chiguata. This refers to low-elevation springs, and this has been clarified in the revised manuscript. Instrument and standards used for tritium analysis were added.

Sections 4.1 and 4.2: I suggest merging both sections into a single one as they present very similar and related information. Suggestion for title of new sections: Isotopic composition of precipitation, surface, springs and salar water (i.e., dismiss the times series portion of the titles)

Section 4.2 was kept separate to avoid confusion with groundwater in section 4.1, but the reviewer makes a good point that precip and salar waters are not groundwater either. We don't mind combining them.

L304: cross-referenced subplot 4c) is missing. See comments in Figure 4 below and update accordingly.

Subplot c was removed from the figure, but not the legend. We will fix this.

L308: cross-referenced subplot 5c) is missing. See comments in Figure 5 below and update accordingly.

Subplot c was removed from the figure, but not the legend. We will fix this.

L.373: briefly justify why the use of the Gibbs diagram could be considered robust for the study area groundwater. According to Marandi and Shand (2018), page 211, "It remains possible that recharging waters cross the water table with a chemical composition plotting in the upper right corner of the Gibbs Diagram, perhaps particularly in regions where land uses have caused salinization of soils or other artificial impacts to near surface mineralogy." We propose that saline water from the salar provides recharge (therefore the

salar surface waters plot in the evaporative enrichment field) and then mix with non-saline groundwater flowpaths in the mountain block. We infer that this mixing results in Chiguata springs plotting as a mixture of evaporative enrichment waters and precipitation dominant waters.

L.379: Precipitation dominance for any of the samples as suggested here and in the last line of Figure 8 caption. Please revise and update accordingly.
The Quenales Forest springs and Pichu Pichu springs are supported by high-elevation recharge from snowmelt and possibly rain. They are not supported by recharge from saline groundwater and therefore do not plot in the evaporative enrichment field. They do, however, plot in the precipitation dominance field which is physically realistic and defensible.

Section 4.3 I find it odd that geochemical information on surface waters is not described in the results section, particularly regarding figures 9 and 10. Please revise the whole section and describe important results regarding surface waters.
The primary purpose of this project was to investigate mountain groundwater processes that support low-elevation springs. We can add a brief discussion of the surface waters in the revision if it does not add too much to the length. The main observation here is that river chemistry is similar to nearby springs in their respective locations. This indicates groundwater discharge is supporting river flow.

L.418: how was it identified that surface waters were not evaporated? Please mention this in results sections and cross-reference a figure or table to support this observation.
We were referring to the wet season laguna salinas surface water presented in Fig 5.
L.430: enhance local evaporation
L.442: both isotopes actually
L.445: Please show the isotopic composition of Laguna Salinas surface water during the dry and rainy season in the figures.
They are.
L.449: add elevation of Tacune mountains
L.478: cross-reference Fig. 6
L.482: relative to surface-
L.508: please cross-reference Fig. 7
L.548: it is
Table 1: assign a code to precipitation sampling site and add it to the table. Also, specify the period of rainwater sampling, it seems it was January-March 2019 according to the text.
Table 2: as in Table 1, please show clearly which sample sites correspond to the Characato and Chiguate districs.
We can do this.
Figure 2: Suggest to use a topographical map instead so that the elevation differences are more easily visualized. Also, it would be very useful for the reader if the area shown in figure 3 would be marked in this map for reference. It would also be very helpful to show

the different water types samples in different colors for reference in the legend of the figure. Also include this in the caption: "Names of the sampling sites are shown in Table 1 for reference".

These are nice suggestions. The other reviewer suggested we include the faults and using different symbols for the types of samples would be helpful. Adding topography might make it too busy though.

Figure 4: subplot c) is missing. Either add the subplot or update the caption of the figure accordingly. Also, please mark the dry season Laguna Salinas surface water samples in a) for reference.

Subplot c was removed from the figure, but not the legend. Will fix this.

Figure 5: the figure has very low quality, please update it to meet publication standards. Subplot c) is missing. Either add the subplot or update the caption of the figure accordingly.

Subplot c was removed from the figure, but not the legend. Will fix this.

 Also, please mark the dry season Laguna Salinas surface water samples in a) for reference.

Figure 6: add the IDs of the sampling sites as those are also used in the manuscript.

Good idea.

Figure 10: quality of the Figure seems to be low. Please improve it.

The quality can be improved

Technical issues – all technical issues can be addressed in the revised manuscript

L67: have formed

L74-77: Very long sentence, difficult to understand. Please rewrite.

L75: suggest using the term tracer instead of component here and in the whole manuscript.

L116: study area

L130: the population of the capital city

L144: 6.7 Ma ago?

L217-219: sentence is difficult to read, please rewrite.

L245: odd sentence in caption of Figure 3. Revise.

L256: six smaller high-elevation

L265: …in Characato was used to collect rainwater. Also, use the same number of decimals as in Table 1.

L286: were obtained

L.345: spring waters instead of springs

Figure 11: I think it is better to keep using the a) and b) type of cross-reference for the subplots for consistency throughout the manuscript, instead of the current top/bottom.

The other reviewer suggested combining this into 1 panel.

References:

Albero, M.C. and Panarello, H.O.: Tritium and stable isotopes in precipitation water in South America, 1981

Marandi, A. and Shand, P.: Groundwater chemistry and the Gibbs Diagram, Applied Geochemistry, 97, 209–212, https://doi.org/10.1016/j.apgeochem.2018.07.009, 2018.

---

## Author Response (AR1)

**Reviewer 1 comments (response in red)**

**General Comments**

This is an interesting study that helps inform our understanding of recharge, groundwater flow, and spring origins. However, it needs attention before it is suitable for publication.

Firstly, the paper needs better focus. Some sections (e.g. the discussion of geochemical processes) do not seem to inform the overall story and are a distraction. The description of the study area is long and some of the information does not seem to be relevant. There are also descriptive sections in the discussion and a tendency to repeat information. A more succinct tightly-written paper would be easier to follow and have more impact.

More importantly, in several places the discussion of processes is not convincing (inc. the geochemical processes, groundwater residence times, and lapse rates). The explanations tend to be long and are sometimes not consistent or are overly speculative. The central point that the springs are recharged at altitude and the water follow different flowpaths appears reasonable, but this sometimes is lost.

Finally, the Conclusions are mainly parochial . Some idea of the broader significance of this study or comparisons with similar environments would give the paper a broader appeal.

I hope that these comments are useful.

We thank for the reviewer for the thoughtful feedback and suggestions for making the manuscript more concise.

**Specific Comments**

*Abstract*

The abstract gives a good idea of the major conclusions of the paper. However, it needs more focus on the results and conclusions rather than the aims. You should report a few key values in the Abstract as qualitative descriptors ("stable", "higher" etc) do not convey much specific information.

The review makes good suggestions. We have edited the abstract to include more results.

*Introduction*

The introduction is clear and well structured. The first paragraph (lines 40-55) would benefit from a few extra details if these are available, specifically:

- Are the impacts on groundwater resources quantified? Groundwater resources in this region are relatively poorly quantified. There are substantial knowledge gaps with respect to hydrogeology in the region and our paper attempts to address a subset of these knowledge gaps. We added the following sentence to cite some literature making this point. *"The importance of MBR on groundwater resources in the Andes and how they might respond to changing precipitation and temperature and melting glaciers are poorly understood (Somers et al., 2018; Somers et al., 2019)."*

- Likewise are there estimates for how much the recharge may decline? No, this is difficult without 1) some baseline assessment on how much recharge occurs historically in this region, 2) some assessment on the sources of recharge (provided in our paper), and 3) some information on how regional climate change will impact the source of recharge (e.g., changes in rainfall, changes in high-elevation snowfall, loss of alpine glacial cover). These data are lacking and these projections are outside the scope of our study.

This would help with understanding the context of this study.

Lines 57-65. Provide some references for this material. References added.

Lines 73-79. This is only true is there is no other mechanism for exporting Cl. In some saline lakes Cl is lost via salt deposits being eroded by the wind (deflation). Is that the case here? The reviewer's point is noted and we added wind erosion as a potential mechanism to remove salts from the salar. The studies that we cite in this introductory section have not come to that conclusion though. *"By using the ratio of the total mass of chloride (assumed to be a conservative tracer and assuming insignificant wind erosion) in the brine lakes to the annual input flux to estimate the tracer's residence time, Risacher et al. (2003) determined that the residence times of Cl⁻ in Chilean brine lakes were short (few to hundreds of years) in a flow-through steady-state condition."*

If the lakes do recharge the local groundwater system, I would expect there to be some evidence of that (shallow high salinity groundwater around the lakes) – any evidence of that?

With respect to high-salinity shallow groundwater in the salar, wells are sparse in this area and we do not have access to wells used by mining companies. We edited section

5.4 to discuss our interpretation of the 1 well in the salar we do have chemistry on (INK) and inferences from the sediment composition.

*Study Area*

This is comprehensive, but in places the descriptions are lengthy.

- There is some repetition with the introduction (eg lines 119-129). The information provided in this section is specific to our study area and highlights the importance of La Bedoya spring and the intensity of agricultural and mining activities in this region. We condensed the material to these main points.

- Some of the details of the geological history (lines 137-148) seem superfluous. There are only 2 sentences that reference the geologic history, but we trimmed slightly.

- The climate description (Section 2.3) could also be more succinct – the important details are a bit lost in the narrative. This is now section 2.2. We cut the description of recent trends which are not particularly relevant to this study and condense that idea in the introduction motivation.

- Again with Section 2.4, how much of this detail is really necessary

  This is now section 2.3 We focus on describing the specific low and high permeability geologic units and the faults located in the region. We describe the salar but save the geochemical description of the lacustrine deposits for the discussion when they are most relevant to the reader.

The lengthy descriptions mean that it is not always clear what the important points are. Try to give more focus to what is important for the study rather than trying to cover everything. We believe specific comments related to this theme are addressed below.

The order is also not intuitive – Sections 2.2 and 2.4 both deal in part with the Geology and Geomorphology. It would be better to merge these two (or at least make them sequential) and to present the material in a large- to smaller scale order. So the descriptions of the large-scale geology (lines 200-207) would be better presented before the description of the basin (Section 2.2).

The order for section 2 is now: 2.1 water uses; 2.2 climate; 2.3 geological description; 2.4 watershed description and sampling locations. We hope this improves readability.

Figures 2 & 3 can be merged – they are maps of the same area. That way it would be easier to see the relationships between the samples and the faults. Also, it is not currently clear from Fig. 2 what is sampled (springs, surface water etc) without referring to the Table – make the symbol for each different.

These figures are now combined with different symbols. We hope the reviewer likes this result.

*Methods*

Here also there are some diversions (eg Lines 261-265) that are distracting and do not belong in the methods. If this information is important, then it belongs in the study area section.

The additional descriptions of sampling locations were moved to the study area section.

Lines 270-272. Really it is only SpC that is measured (This is not correct. It is only the electrical conductivity that is measured. Specific conductivity and TDS are calculated from the electrical conductivity), the TDS is just calculated from the SpC using an assumed conversion. If you have a full suite of ions (which it looks like you do), you can calculate TDS as the sum of the ions. If you stick with the TDS estimated from SpC, you need to note that and provide the conversion.

This text was added to section 3.1. "Electrical conductivity (EC) was measured in the field using a YSI Professional Plus (Quatro) multi-parameter probe. The YSI probe calculates 1) specific conductivity (SpC, $\mu$S cm$^{-1}$) from: SpC = EC * 1.91; where 1.91 is the temperature coefficient at 25°C, and 2) total dissolved solids (TDS, ppm) from: TDS = EC * 0.65.. (Supplemental Table 1)."

Section 3.2. A few more details on the Tritium procedure (enrichment and equipment) needed

This text was added to section 3.2. "Samples of one litre were collected for $^{3}$H in the field, stored in high density polyethylene bottles with leak free caps and were not exposed to indoor air. Samples PPI, OJO, BED, INK, and UBI, as well as a one-time 1L sample of rain within the city of Arequipa during Feb 2020, were analyzed for $^{3}$H at the University of Miami, Tritium Lab. Water samples were distilled, enriched via an electrolyzed water bath for 10 to14 days, reduced over hot magnesium metal, and transferred to the counter for 6 to 20 hours. Backgrounds are set by comparison to tritium dead petroleum and deep Florida Aquifer waters. Efficiency is determined by preparation of the NIST SRM#4926 standard water. The reported accuracy and precision for ultralow activity electrolytic enrichment through volume reduction is 0.1 TU. "

Section 3.3. Need to report precision for the major ions

The text below was added and SI Table 1 now includes detection limits and percent recovery precision estimates for each ion. It feels like too much to report them all in the main text.

"Precision for major ions varied by ion but was < 6 mg/L for everything reported here. Additional precision details can be found in Supplemental Table 1.  Boron and Lithium concentrations were determined using ICP-OES at Purdue University (iCAP 7400, Thermo Scientific, China) with dual plasma view, equipped with Qtegra ISDS software. Concentration standards for both ranged from 2 to 2,000 ppb and had correlation coefficients of 0.9959 for B and 0.9948 for Li. The limits of detection for B and Li were 1.1 and 0.1 ppb and the limits of quantification were 3.6 and 0.2 ppb, respectively. The accuracies for both elements were above 90%. Values above 2,000 ppb were reported as above the maximum calibration range."

*Results*

As explained below, this section loses focus in places and it is not always clear what the important points are. You need to be clearer as to what information is important for this study and concentrate on that. The figures need improvements -duplicating the figures just to show the two Salinas samples is excessive. It is clear that these are evaporated surface waters and you do not say much more about them than that. In that case just omit them from the figures and note that in the caption.,

H isotope ratios should have no decimal places to be consistent with the quoted precision (O is fine with one decimal place). Fixed

Lines 314-317. Report the low 3H activities as bd without the +/- (which have no meaning for 3H below the lower detection limit). If the precision is 0.1 TU (Section 3.2) then the +/- of 0.04 for BED is overoptimistic. Given that this is very close to the lower detection limit, how sure are you that this is real. Given that you sampled the springs at their surface outlets, could it just be a small amount of modern water (eg recent rainfall) mixed in with the spring water?

We state in the manuscript that water samples at or below the detection limit are tritium dead. Samples with 3H values that were below the detection limit are reported as tritium dead and lab-reported numeric values were removed from the manuscript and SI Table 1.

Figure 4. The two diagrams are a little confusing as they present the same data. If you feel that you need 4b to show the detail, you could make 4a in inset with just the Salinas surface water on it. Also need to reference the LMWL in the caption.

These are combined into Fig 3.

Figure 5. Again, you do not need both figures – you can just use Fig. 5b and explain that you have omitted the Salinas waters. It would be useful to have Fig. 4 & 5 as a single figure as it would show the relevant stable isotope data in one place.

We took the reviewer's suggestion and these are now combined in Fig 3.

Section 4.3 Now section 4.2

This section lacks focus. What is important here (the processes or the differences in chemistry). The discussion of the processes using the Gibbs Diagram is not very convincing and could be done better. however, consider whether that is important. If the important point is that the waters have different geochemistry and so follow different flow paths, then just show that (the Piper and/or a couple of bivariate plots like Fig. 10, and a brief description would suffice). The discussion of processes is not very clear and may not be necessary.

The Gibbs Diagram in particular is not very informative in determining processes and you have the data to do that more rigorously. Waters dominated by evapotranspiration have Cl/Br and Na/Cl ratios close to those of rainfall (which you can probably estimate). Extensive rock weathering produces high cation/Cl ratios while halite dissolution produces very high Cl/Br ratios. A few bivariate plots (eg Na/Cl vs. TDS and Cl/Br vs. TDS) would show that much better than the approach that you are currently using. There is extensive literature on this (eg numerous groundwater papers by Mike Edmunds).

The Gibbs diagrams were used to show that flowpath groups fall within the expected processes within the diagram (eg. Salar waters within evaporation and bedrock springs within rock weathering). We also reference the paper by Marandi and Shand (2018) which uses an example from Edmunds et al. (1987) to show that groundwaters which plot in the evaporative enrichment region of the Gibbs diagram may not have experienced dissolution of evaporites. In the case of Edmunds et al. (1987), the groundwaters evolved toward this Na-Cl type water due, in part, to mixing with saline connate groundwater. Thus, an argument can be made against the Gibbs diagram that false evaporation processes can arise in geologic settings where marine deposits and saline groundwater are present. In our study area there are no ancient marine deposits that would conflate our interpretation of the salar waters as originating from

evaporative processes. The Gibbs diagram clearly shows that the low-elevation spring samples which appear to be receiving some mixture of high-elevation high-salinity recharge fall along a proposed mixing line with this subspace. This does not appear as clearly in the plots suggested by reviewers (provided below). The Cl/Br plots are also problematic because few samples had detectable levels of Br and so this data set is much smaller than that shown in the Na/Cl and Gibbs diagram.

In addition, Marandi and Shand (2018) state in their critque of Gibbs DIagrams that, "It remains possible that recharging waters cross the water table with a chemical composition plotting in the upper right corner of the Gibbs Diagram, perhaps particularly in regions where land uses have caused salinization of soils or other artificial impacts to near surface mineralogy." This is the exact condition which we propose is happening in the connection between saline recharge from the salar and the Chiguata springs. Saline recharge would plot in the evaporative enrichment field where the Laguna Salinas surface waters plot. This saline recharge then mixes with non-saline groundwater flowpaths in the mountain block that were recharged by high-elevation snow and/or rain. Thus, Chiguata springs plot between the fields for evaporative enrichment and precipitation dominance. In comparison, springs such as the Quenales Forest springs and Pichu Pichu springs, which are supported by high-elevation recharge from snowmelt and/or rain, plot near the precipitation dominance field for anions and rock-weathering for cations. Their placement in these fields make physical sense.

We restructured and streamlined this section to make these points clear. We are not convinced that adding other bivariate plots will necessarily change this story. For example, we plotted Cl/Br mass ratios and these essentially show the same result. Saline recharge has high Cl/Br and relatively low TDS because high-elevation springs we sampled flow into the salar and the salar refills during the rainy season with dilute runoff from surrounding mountains. As this saline recharge mixes with other non-saline groundwater flowpaths, the Cl/Br ratios tend to decrease while the TDS increases due to rock-weathering reactions. This interpretation is not different from that inferred from the Gibbs Diagram.

[Figure]

Lines 386-394. I presume that this also shows up in the other parameters? I'm not sure that you have enough data to do a PCA or cluster analysis but you should make the point with the other parameters.

We do not think it's appropriate to use a cluster analysis on this dataset. Figures 8 and 10 supports this interpretation as well. We edited the discussion to describe mixing along the multiple groundwater flow paths.

*Discussion*

The Discussion covers a range of topics but there are a number of potential inconsistencies and unclear explanations. This is the most important part of the paper, so more clarity would help. I also suggest that you add a couple of sentences to the start of the Discussion as a guide to what you will be dealing with.

Sections 5.1.1 & 2.

The discussion regrading the lapse rate (lines 415-434) needs to be clearer. While it is true that springs can have stable isotope ratios that vary with altitude that is mainly the case where they are recharged close to where they discharge. In the case of your springs, you make the case (Section 5.1.2) that they are recharged at high altitudes. That interpretation is reasonable. However, the way that this discussion is presented is to set up the idea that the springs should vary with altitude (line 416) and then point out that that is not the case and then interpret the data in terms of recharge altitude in Section 5.1.2.

We edited the framing to make it easier on the reader.

Much of Section 5.1.1 (the correlation with altitude etc) contains observations that should be part of Section 4 – some of it is in there already and it is just repeated here.

We do not think any new results are presented in section 5.1.1 other than 1 sentence reminding the reader that the isotopic composition of rain at the lower elevation matches the spring water at the higher elevation. We cut that. The rest of this section provides discussion and context for the observations reported in Section 4.

The magnitude of the lapse rate.  If the springs are recharged at higher altitudes then you can't use them to estimate the lapse rate. In that case, your lapse rate should be based on the surface water samples.

We agree with the reviewer's point that springs recharged at higher elevations will be more isotopically depleted than precipitation at the same elevation as the spring emergence unless the flowpaths are very short or storage is small. The curious thing in this case is that the springs at a HIGHER elevation were isotopicaly similar to precipitation at a LOWER elevation in 2019. Since we do not claim that lower elevation rain is recharging higher elevation springs (against gravity), we can conclude that precipitation at the higher elevation spring recharge area is similar to the lower elevation precipitation we observed.

We revised this section 5.1 to center the discussion on the similarity between the spring values in the queñuales forest and Characato to infer they have similar zones of recharge at or above 4000 masl. The discussion of lapse rates with elevation are moved later in the section and the emphasis reduced. We would be open to deleting the paragraph on lapse rates entirely if the reviewers think that's better.

However, those data may not be suitable, specifically:

- Rainfall sampling is referenced to an unpublished study. It is not clear how many samples this represents and the duration of the rainfall record. Given the likely variability of rainfall isotope values, you ideally would have a multi-year weighted average value, but is this the case?

This is true. Precipitation collection was initiated for this project and included daily rain collections during the 2019 rainy season (effectively 1 year because there is only trace rain in other months). In the revision, we can now also include additional data generated after this manuscript submission that shows 2020 precipitation collected at both Characato and Chiguata were less negative than the 2019 precipitation collected at Characato.

Characato 2019 (-9.12,-63.51); Chiguata 2020 (-6.3,-38.1) ; Characato 2020 (-5.5, -31)

- Snow is probably mainly winter precipitation and is difficult to use with samples that represent long-term averages.

We are unclear what the reviewer's point is here. The snow on Pichu Pichu occurs during the summer. Summer is the only season with measurable precipitation with rain at lower elevations and snow at higher elevations. Seasonal bias is not a complication in this system. We added an SI figures section with average maximum and minimum monthly temperatures and average monthly precipitation at 3 different SENAHMI meteorological stations locations near Characato, Chiguata, and Lagunas Salinas.

- The surface water samples seem to be partially fed by spring water (Section 5.2). If those springs were recharged at high altitudes then using river water to calculate lapse rate is possibly not valid as it is not capturing only rainfall at the altitude where you sample it.

We did not use the surface waters to calculate the lapse rate in the original submission or in the revision.

- A similar concern would apply to any rivers that flow from high to low altitudes and thus mix rainfall from a variety of altitudes

Agreed. See earlier comments.

- Even if the rivers are mainly fed by local rainfall, their stable isotope values are likely to vary seasonally (as you discuss in Section 5.2) and so again are difficult to use in this way.

Agreed. See earlier comments.

I'm not convinced that you can determine the lapse rate with the data that you have. If you are going to include this discussion, it needs to be more convincing. Otherwise you may be able to estimate it using other studies?

The main goal is to estimate the recharge elevation of the lower-elevation Characato springs. These lower elevation springs are isotopically similar to the mid-elevation Quenalaes springs. We interpret the recharge zone of the low elevation springs as the same elevation of the Quenalaes spring recharge. This is the most important observation. The discussion of the inferred precipitation isotopic lapse rate may have created a distraction for the reader and its emphasis has been reduced.

Spring recharge elevation. This seems broadly correct; however, it becomes more doubtful if there are palaeowaters in the basin (lines 463-465). Discharge of paleowaters into surface water bodies also complicates the lapse rate calculations. Is there anyway to test this idea? While you do not have radiocarbon data are there examples of palaeowaters in analogous settings or examples of nearby springs for which there are better residence time calculations?

We do not have evidence of paleowater discharge from this basin, but since the region has a glacial history, we cannot absolutely rule it out without additional radiogenic tracer data. We thought it best to mention this possibility.

Section 5.3 also needs attention.

- It is not clear where the 3H activity of "young aquifers" of 2.9 TU comes from

This came from recharge-weighted well-mixed aquifer mixing models informed using time series from Albero & Panarello (1981) - this includes a compilation of time-series from South America. High-elevation snow was measured in this study and it had a tritium activity of 2.2 TU. A tritium activity of 2.9 TU is not beyond the realm of possibilities based on our measured data. In fact, we commonly find that

tritium activities decrease with decreasing elevation (with inferred increasing flowpath length, degree of mixing, and storage). Manciati et al (2021) show a mean precipitation activity of 3.0 TU near Quito, Ecuador which is admittedly north of our study site, but not disparate from our measurement on Pichu Pichu Volcano snow. This statement has been removed.

"While it is not possible to provide an accurate residence time based solely on $^3$H data for these springs, we infer that the residence times of springs in the study area are likely greater than 60 years."

- The definition of fossil water as being >60 years is largely a northern hemisphere viewpoint as the higher 3H bomb pulse waters are still detectable in groundwater. This is not the case in the southern hemisphere where the bomb-pulse tritium has decayed back to natural levels (e.g., Morgenstern, U., Stewart, M.K., Stenger, R., 2010. Dating of streamwater using tritium in a post nuclear bomb pulse world: Continuous variation of mean transit time with streamflow. Hydrology and Earth System Sciences, 14, 2289-2301; Tadros, C.V., Hughes, C.E., Crawford, J., Hollins, S.E., Chisari, R., 2014. Tritium in Australian precipitation: A 50 year record. Journal of Hydrology, 513, 262-273).

  The modern precipitation samples we analyzed found values between 1.4 and 2.2 TU. This shows that new inputs in this region are not yet 3H dead. The word 'fossil' does not appear in the manuscript now because it is ambiguous. Tritium-dead groundwaters are not unique to the southern hemisphere; this is increasingly a problem in the northern hemisphere. "Tritium dead" is admittedly an ambiguous term, as we state in the manuscript. We do use the interpretation provided by Moran et al. (2019) that 3H dead waters in this region are greater than 60 years old. "While it is not possible to provide an accurate residence time based solely on $^3$H data for these springs, we infer that the residence times of springs in the study area are likely greater than 60 years."

  Rather than use tritium data from Australia, which is well documented, we wanted to place our tritium data in the context of the tritium breakthrough in South America. We specifically state in the manuscript that, "The atmospheric breakthrough of $^3$H associated with nuclear weapons testing in the 1950s and 1960s for South America peaked at approximately 60 TU in Cuiabá, Brazil in 1965 while equatorial monitoring stations typically had $^3$H breakthrough activities less than 40 TU (Albero & Panarello, 1981)."

- The residence time of 300 years seems arbitrary. Presumably it is based on mixing at the top of the aquifer but you have a fractured flow system that is likely to behave very differently.

This discussion has been removed in the revision. This estimate was based on recharge-weighted well-mixed aquifer models with continuous precipitation input that were informed using time series from Albero & Panarello (1981) and some back-of-the-envelope calculations of specific discharge. The mixing models have high uncertainty because these models are sensitive to the fraction of recharge that is used to weight the tritium annually. We think that these springs likely have residence times falling between the ranges for tritium and radiocarbon. The flow calculations are uncertain because the region is data poor and we do not have well-constrained hydraulic parameters. Therefore, we will simply state that the groundwaters are older than 60 years until additional data become available.

As you have only three tritium measurements (all of which are close to or below detection) and you do not have a good idea of the rainfall values (2.5 to 10 TU is a large range), there is little quantitative that you can say here and this section is not that informative. I'd just make a case for the water being at least a few decades old in Section 5.4.

We believe the reviewer is suggesting that we remove section 5.3 entirely. We chose to retain the section because it provides 2 key elements of interpretation. "For context, Moran et al. (2019) sampled high-elevation lakes and lagoons that had $^3$H activities of approximately 1.0 TU near the Salar de Atacama in Chile. In comparison, the two high elevation springs in Laguna Salinas basin (INK and UBI) were 3H dead indicating long or slow flowpaths in local high elevation recharge and groundwater mixing below the salar.  The two lower elevation springs (OJO and BED) were also $^3$H dead. These results are consistent with the nearly constant stable isotope values of the Chiguata and Characato district springs, showing stable, relatively groundwaters."

Section 5.4

Lines 517-520. I'm not sure that I'd expect Cl to increase along flow paths. To do so implies that Cl needs to be added from the rock matrix as evapotranspiration is a surface process. That will only occur if there is halite in the rocks (which is not that common). This concept does appear in many textbooks but the supposed process is never really explained.

Cl concentrations should only be invariable along flow paths if: 1) piston-flow processes are invoked and groundwater doesn't mix with other flowpaths, or 2) if Cl is not removed or added by dilution or biological processes. We propose that mixing occurs

at this large spatial scale within a fractured mountain block. We cite Bresciani et al. 2018 for this.

Lines 517-525. You look to have measured Br. Cl/Br ratios will readily determine whether you have halite dissolution (Cartwright, I., Weaver, T.R., Fifield, L.K., 2006. Cl/Br ratios and environmental isotopes as indicators of recharge variability and groundwater flow: An example from the southeast Murray Basin, Australia. Chemical Geology 231, 38-56). You may not need to speculate here.

Br was not detected in all samples, but below is a plot of samples with Br measurements from Chiguata and the springs feeding into Lagunas Salinas. The plot below is consistent with halite dissolution influence in the Chiguata springs. The Lagunas Salinas springs (UBI and MOQ) are sampled before they interact with the salar water. Salar surface water in the wet season has Cl = 6350 mg/L and Cl/Br = 872. This would plot off to the lower right and not indicate halite dissolution influence.

[Figure]

[Figure]

Section 5.5.

The first paragraph (lines 569-574) repeats the previous section and is not needed.
fixed

Figure 11 only needs one panel as you can show all three flow paths without confusion.

We have merged the information in this figure in the revision.

*Conclusions*

Again, there is some repetition here. Instead of repeating the specific findings, which you cover in Section 5, try to outline the general points. However, you should explain the general importance of the study or compare it with similar studies elsewhere. This will make the paper appeal to a wider readership.

We shortened the summary information and focus on the broader impacts.

Two general points we can emphasize are:

1) Results from our study show evidence of this high elevation closed basin salar is hydrologically connected to lower elevation springs. We place this in the context of

other salar studies in the Andes and conclude that not all salars are evaporation pans, but as other studies have found, some discharge from the basin enters regional groundwater.

2) This study provides perspective on the long-term stability of groundwater in this arid region. The tritium data as well as repeat discharge and stable isotope measurements supports long-flow paths with residence times of water being at least a few decades old if not longer. In the face of climatic warming these data suggest that regional groundwater may provide a stable water resource over the next few decades.

References:

Albero, M.C. and Panarello, H.O.: Tritium and stable isotopes in precipitation water in South America, 1981

Bresciani, E., Cranswick, R. H., Banks, E. W., Batlle-Aguilar, J., Cook, P. G., and Batelaan, O.: Using hydraulic head, chloride and electrical conductivity data to distinguish between mountain-front and mountain-block recharge to basin aquifers, Hydrol. Earth Syst. Sci., 22, 1629–1648, https://doi.org/10.5194/hess-22-1629-2018, 2018.

Edmunds, W. M., Cook, J. M., Darling, W. G., Kinniburgh, D. G., Miles, D. L., Bath, A. H., Morgan-Jones, M., and Andrews, J. N.: Baseline geochemical conditions in the Chalk aquifer, Berkshire, U.K.: a basis for groundwater quality management, Applied Geochemistry, 2, 251–274, https://doi.org/10.1016/0883-2927(87)90042-4, 1987.

Manciati, C., Taupin, J. D., Patris, N., Leduc, C., and Casiot, C.: Diverging Water Ages Inferred From Hydrodynamics, Hydrochemical and Isotopic Tracers in a Tropical Andean Volcano-Sedimentary Confined Aquifer System, Front. Water, 3, 597641, https://doi.org/10.3389/frwa.2021.597641, 2021.

Marandi, A. and Shand, P.: Groundwater chemistry and the Gibbs Diagram, Applied Geochemistry, 97, 209–212, https://doi.org/10.1016/j.apgeochem.2018.07.009, 2018.

Moran, B. J., Boutt, D. F., and Munk, L. A.: Stable and Radioisotope Systematics Reveal Fossil Water as Fundamental Characteristic of Arid Orogenic-Scale Groundwater Systems, Water Resour. Res., 55, 11295–11315, https://doi.org/10.1029/2019WR026386, 2019.

**Reviewer 2 comments (response in red)**

The paper "Evidence for high-elevation salar recharge and interbasin groundwater flow in the Western Cordillera of the Peruvian Andes" by Alvarez-Campos presents a multi tracer (isotopic and geochemical) assessment on the influence of groundwater flowpaths from a close basin salar in spring water upwelling at lower elevations that supply water to the city of Arequipa, Peru. Overall, I find the paper well-structured and clearly written and the findings generally well supported by the presented data and analysis. My major concern relates to the insufficient description of sampling collection and laboratory analyses. Given the relevance of the paper for the management of water resources in the region of study, I consider it is suitable for publication in HESS after some points described below are implemented in the manuscript.

We thank the reviewer for highlighting methodological details that were missing and have added them to the manuscript.

Major comments:
L111-113: tritium and residence time come as a surprise for the reader. I suggest adding a few statements or a short paragraph to the introduction mentioning the value of tritium in the context of the study, highlighting particularly research on the study region for similar purposes.
We added a few sentences about the use of tritium in other studies in the region to the introduction. A expanded explanation of the application was added in the methods section 3.2. We felt adding a lot of methodological detail to the introduction could be distracting for the reader.

Section 3.1. It would be helpful to include the elevation of the sites for reference in this section so the reader does not need to check Table 1 many times. Also, it is confusing that the authors sometimes mentioned only the names of the sites, other times only the sites IDs (presented in Table 1), and others, both names and IDS. I strongly suggest to homogenize this in the whole manuscript, figures and tables for consistency and clarity (i.e., this issue is common in this and the rest of the paper sections).
This is a helpful suggestion to improve readability of the manuscript. Elevations were added to section 3.1 and we tried to consistently use the sample IDs throughout.

L260: describe how snow was sampled.
Detail on snow sampling were added to section 3.1.

L261: report the period and frequency of rainwater sampling. If not collected throughout the whole study period, indicate why. I strongly suggest to give a name to the precipitation water sampling in Table 1 and use it in the whole manuscript. Also, show it and add it to the legend in all relevant maps.

Rain was sampled over the wet season January-March. The rest of the year there is no precipitation to sample. We included this clarification and were able to add additional rain isotope data from another site near Chiguata in 2020. These are included in Table 1 and added to Figure 2.

L267-272: the description of water sampling collection is quite incomplete and requires substantial improvement. Some of the main issues are: how was water from river and springs samples? How were samples collected for stable isotope analysis stored to avoid fractionation by evaporation? How was rainwater sampled to assure evaporative fractionation did not affect the water samples? What sits were samples for tritium, specify? Report the made, model, and accuracy of devices used to measure physico-chemical parameters in situ and how often and how they were calibrated. Please update the paragraph with this and other relevant information that might be missing.
We detail in the methods isotope section 3.1 how samples were collected, storages and sealed to avoid evaporation. We have added further details on rainwater collection and evaporation minimization. Samples for tritium are listed now in the methods section 3.2 and indicated in Table 1. In situ calibration and instrumentation data are included now for the Oakton pH meter and daily sensor calibration in the field.

L286: I am puzzled about the construction of the LMWL using data for 3 months only. In section 2.3, it is mentioned that the very dry winter occurs between June and August, however, it is not clear if precipitation during those months is at all nonexistent, or just very little compared to the wet summer monsoon one (November to April). Even for the latter, using an isotopic dataset from January through March 2019 might not be entirely representative of the local conditions. I strongly suggest the authors to include a time series of precipitation during the study period in the paper for reference, and discuss if and how the limitation of the available isotopic dataset could influence their findings. Showing the precipitation amount data could also help to link their findings about the influence of modern day recharge on their findings and the developed conceptual model.
It only rains a few months out of the year, during the summer (~Dec-Mar). Our 2019 precipitation sampling was effectively an annual summary. We now have daily collections for 2020 as well from Characato and a new station in Chiguata. For the reviewer's reference, the LMWL calculated for the 2019 wet season was y = 8.01x + 8.40. Including 2020 precipitation, the fit was very similar, y = 8.05x + 10.91. We have added this new data from 2020 which shows substantial interannual variability in precipitation isotopes at this location. We have included SENAHMIi data of monthly mean precipitation at 3 locations: Characato, Chiguata, and near Lagunas Salinas in SI Fig 1 showing that precipitation is nearly non-existent outside of the wet season. The reviewer should note that we do not base any of the conclusions or analysis in this paper on the LMWL. Nor are recharge elevations inferred by the year or 2 of precipitation collected in this study. Most conclusions are based on relative differences of the springs and salar surface water. The 2019-2020 precipitation data is only provided for general context. A separate manuscript is

submitted that analyzes the daily precipitation water isotope data from an atmospheric process perspective. Adding it here would add unnecessary detail to an already lengthy paper.

Section 3.3: there is very little information about the chemical analysis. Please report standards, calibration curves, detection limits, etc. used for the analysis of anions and cations. Also report QA/QC procedures to secure high quality of the produced data. Summary statements of precision were included in section 3.3. Including calibration curves for every species in the manuscript would not be practical. We have added estimates of uncertainty to SI Table 1.

 "The detection limit in mg/L for each solute/analyte is shown in brackets underneath the solute/analyte concentration in mg/L.
The analytical uncertainty is reported in parentheses underneath the detection limit in brackets. The uncertainty is reported as (100 - percent of standard recovered in measurement). For example, if a 50 mg/L standard is sampled prior to actually measuring the sample, and the standard measures 50 mg/L, then the percent recovery is 100% (equivalent to 0% uncertainty). Maximum deviations from the standard are reported to be conservative."

L.346-350: I strongly suggest to show the data supporting these statements (i.e., similarity between spring and surface waters). One option is to have a subplot in Figure 6 showing the springs' isotopic compositions. It would also be good to include the isotopic composition of precipitation in such a plot (e.g., adding a third panel, or plotting together with the springs and surface water isotopic fingerprints?)
We edited (now) Fig 4 to aid in the comparison that the review suggests. The average spring water compositions were added as dashed horizontal lines with max/min ranges reported in the figure caption. We believe this will help the reader make the comparison between springs and rivers now.

L.447 and L.502: how do the authors infer that residence time should be several hundred years old? If anything, based on the Tritium dead results, one could say that groundwater is older than ca. 60 years based on the 1960s bombings. However, without further evidence, saying that water is of certain age seems arbitrary and could be misleading. The authors might be right, but further discussion is needed to justify their statement. Otherwise, please recognize the limitations of the presented dataset and do not speculate about water aging. Based on this comment, I strongly suggest the title of section 5.3 is updated to "Insights into groundwater age" or something similar since results presented are not conclusive.
This is based on recharge-weighted well-mixed aquifer models that were informed using time series from Albero & Panarello (1981) and some back-of-the-envelope calculations of specific discharge. The mixing models have high uncertainty because these models are sensitive to the fraction of recharge that is used to weight the tritium

annually. We think that these springs likely have residence times falling between the ranges for tritium and radiocarbon. The flow calculations are uncertain because the region is data poor and we do not have well-constrained hydraulic parameters. Therefore, we will simply state that the groundwaters are older than 60 years until additional data become available.

Light/heavy versus enriched/depleted: throughout the manuscript, the authors use these terms interchangeably. I strongly suggest the authors to avoid using the terms light/er when referring to depleted isotopic compositions to avoid confusion with the commonly used isotopic terminology of light (more abundant) versus heavy (less abundant) isotope ratios. Please revise the whole manuscript to make changes accordingly.

We made these changes and removed informal usage of light/heavy.

Minor comments:
L50-55: Please support these statements with appropriate references.

"If the amount of precipitation decreases in high-elevation recharge zones, then the amount of effective precipitation (precipitation minus evapotranspiration) potentially available for MBR likewise decreases (Goulden et al., 2012; Goulden and Bales, 2014). In addition, if the land cover changes in the recharge zone (*e.g.,* changes in vegetation type, density, health and/or movement of the treeline), then the MBR will also change. While the effects of MBR reduction may not be immediately felt in the region, they are not inconsequential. Groundwater flow within the mountain block impacts spring flow, headwater streams that originate in the mountain block and subsequently flow across the mountain front, and groundwater wells often located at lower elevations along the mountain front or in adjacent valleys. The importance of MBR on groundwater resources in the Andes and how they might respond to changing precipitation and temperature and melting glaciers are poorly understood (Somers et al., 2018; Somers et al., 2019). "

L65: add references to support the final statement of the paragraph.

"Springs emerging within and down-gradient from the mountain block are important in the high, arid Andes of Peru for a variety of reasons. They provide a source of potable water and can be used for irrigation and recreation, and some have ecological or religious significance (Stensrud, 2019; Sedapar, 2018; Gerencia Regional de Agricultura de Arequipa, 2015). Communities at lower elevations of the western Andes receive very low annual precipitation and rely on surface runoff and groundwater originating from higher elevations (Urrutia and Vuille, 2009). Consequently, identifying groundwater recharge zones and source areas for springs can help us better understand the potential impact of climate change to water resources and better inform water management plans including the protection of perennial springs and their high-elevation recharge zones. This is complicated by the presence of closed-basin salars at high elevations in southern Peru (Juvigné et al., 1997). Some of the runoff that occurs during the wet season is captured by these basins and does not contribute (at least directly) to surface runoff across the mountain front. Furthermore, these basins are not usually

considered a recharge zone (Peña, 2018); however, these salars are local topographic lows and are likely points of groundwater discharge from the surrounding mountains."

L67: this is not true for the whole western South America because i) the northern (tropical) Andes in the north are generally humid and salars are mostly common in areas of the central Andes. Please specify the particular region of the Andes for which this statement applies in the whole manuscript.
We removed this general statement because we think the next sentence contains the qualification the reviewer is looking for.

L96: report elevations of the Laguna and salar
This information was added

L145-147: add references for the statements in these lines
Some of this information is apparent in Fig 2.
"The active Ubinas Volcano (5672 m.a.s.l.) is located northeast from Laguna Salinas in a Quaternary volcanic range (Lebti et al., 2006). Precipitation that falls on Ubinas can drain toward Laguna Salinas or the Río Tambo."

L159: similar to L67, specify the specific region across the Andes for which this statement applies.
We made this clarification. Central Andes

L170: specify which rivers
Added the Rio Majes and Rio Tambo.

L173: from 2018 to ??? please specify
This sentence was modified to specify that this is annual 2018 precipitation data.

L196: report values of the predicted precipitation decrease
The estimates for the 2 studies mentioned were similar. This sentence now appears in the introduction to provide motivation for the study.
"A ~10-30% reduction in precipitation is projected for the western Andeas in the coming century (Minvielle and Garreaud, 2011; Neukom et al., 2015)."

L227-232: Misti and Pichu Picu volcanic complexes are quite relevant for context. It would be super helpful to show them in Figure 3.
We added labels to what is now Fig 2.

L261: report names (or IDs) of the sampled springs
Sample IDs were added in the text and more detail where needed in section 3.1.

L.282: add references for memory effect on isotopic analysis

Added the following reference.

Penna, D., Stenni, B., Šanda, M., Wrede, S., Bogaard, T. A., Michelini, M., Fischer, B. M. C., et al. "Technical note: Evaluation of between-sample memory effects in the analysis of $\delta^2$H and $\delta^{18}$O of water samples measured by laser spectroscopes." *Hydrology and Earth System Sciences* 16, no. 10 (October 31, 2012): 3925–33. https://doi.org/10.5194/hess-16-3925-2012.

L.290-292: six sampling sites are listed here, whereas only four sites were mentioned in section 3.1. Please clarify. Also, please report the instrument and standards used for tritium analysis

The 6 sample site descriptions for tritium collections are now complete in sections 3.1 and 3.2. Instrument and standards used for tritium analysis were added.

Sections 4.1 and 4.2: I suggest merging both sections into a single one as they present very similar and related information. Suggestion for title of new sections: Isotopic composition of precipitation, surface, springs and salar water (i.e., dismiss the times series portion of the titles)

These sections have been combined into what is now section 4.1.

L304: cross-referenced subplot 4c) is missing. See comments in Figure 4 below and update accordingly.

Subplot c was removed from the figure, but not the legend. We fixed this and made other changes to this figure.

L308: cross-referenced subplot 5c) is missing. See comments in Figure 5 below and update accordingly.

Subplot c was removed from the figure, but not the legend. We fixed this and made other changes to this figure.

L.373: briefly justify why the use of the Gibbs diagram could be considered robust for the study area groundwater. According to Marandi and Shand (2018), page 211, "It remains possible that recharging waters cross the water table with a chemical composition plotting in the upper right corner of the Gibbs Diagram, perhaps particularly in regions where land uses have caused salinization of soils or other artificial impacts to near surface mineralogy." We propose that saline water from the salar provides recharge (therefore the salar surface waters plot in the evaporative enrichment field) and then mix with non-saline groundwater flowpaths in the mountain block. We infer that this mixing results in Chiguata springs plotting as a mixture of evaporative enrichment waters and precipitation dominant waters.

Justification text has been added to section 4.3.

L.379: Precipitation dominance for any of the samples as suggested here and in the last line of Figure 8 caption. Please revise and update accordingly.

Revised: "The Quenales Forest springs and Pichu Pichu springs are supported by high-elevation recharge from snowmelt and possibly rain. Plot toward the precipitation dominance area in the cation Gibbs plot and in the rock weathering dominance in the anion Gibbs plot"

Section 4.3 I find it odd that geochemical information on surface waters is not described in the results section, particularly regarding figures 9 and 10. Please revise the whole section and describe important results regarding surface waters.

These results are added to section 4.1 and now Fig 4. This now clearly shows that several of the rivers are supported by groundwater spring discharge, especially during the dry season.

L.418: how was it identified that surface waters were not evaporated? Please mention this in results sections and cross-reference a figure or table to support this observation.

We changed this to wet season salar surface waters. These are plotted in now Fig 3.

L.430: enhance local evaporation fixed
L.442: both isotopes actually fixed
L.445: Please show the isotopic composition of Laguna Salinas surface water during the dry and rainy season in the figures.

Lagunas Salinas surface waters are plotted in now Fig 3. Dry season is in 3a only because they are so enriched. Wet season is in 3a and 3b and plots close to Lagunas Salinas springs
L.449: add elevation of Tacune mountains fixed
L.478: cross-reference Fig. 6 fixed
L.482: relative to surface- fixed
L.508: please cross-reference Fig. 7 fixed
L.548: it is fixed
Table 1: assign a code to precipitation sampling site and add it to the table. Also, specify the period of rainwater sampling, it seems it was January-March 2019 according to the text.
fixed
Table 2: as in Table 1, please show clearly which sample sites correspond to the Characato and Chiguate districs. The table was edited to include district headers.

Figure 2: Suggest to use a topographical map instead so that the elevation differences are more easily visualized. Also, it would be very useful for the reader if the area shown in figure 3 would be marked in this map for reference. It would also be very helpful to show the different water types samples in different colors for reference in the legend of the figure. Also include this in the caption: "Names of the sampling sites are shown in Table 1 for reference".

These are nice suggestions. The other reviewer suggested we include the faults and using different symbols for the types of samples would be helpful. We believe adding topography would make it too busy.

Figure 4: subplot c) is missing. Either add the subplot or update the caption of the figure accordingly. Also, please mark the dry season Laguna Salinas surface water samples in a) for reference.
We have edited this now Fig 3.

Figure 5: the figure has very low quality, please update it to meet publication standards. Subplot c) is missing. Either add the subplot or update the caption of the figure accordingly. Also, please mark the dry season Laguna Salinas surface water samples in a) for reference. fixed
Figure 6: add the IDs of the sampling sites as those are also used in the manuscript.
Good idea. This is now Fig 4 and site IDs are in the legend.
Figure 10: quality of the Figure seems to be low. Please improve it.
This is fixed in now Fig 8.

Technical issues – We believe all the technical issues below have been addressed.
L67: have formed
L74-77: Very long sentence, difficult to understand. Please rewrite.
L75: suggest using the term tracer instead of component here and in the whole manuscript.
L116: study area
L130: the population of the capital city
L144: 6.7 Ma ago?
L217-219: sentence is difficult to read, please rewrite.
L245: odd sentence in caption of Figure 3. Revise.
L256: six smaller high-elevation
L265: …in Characato was used to collect rainwater. Also, use the same number of decimals as in Table 1.
L286: were obtained
L.345: spring waters instead of springs
Figure 11: I think it is better to keep using the a) and b) type of cross-reference for the subplots for consistency throughout the manuscript, instead of the current top/bottom.
The other reviewer suggested combining this into 1 panel.

References:

Albero, M.C. and Panarello, H.O.: Tritium and stable isotopes in precipitation water in South America, 1981

Marandi, A. and Shand, P.: Groundwater chemistry and the Gibbs Diagram, Applied Geochemistry, 97, 209–212, https://doi.org/10.1016/j.apgeochem.2018.07.009, 2018.